# A lncRNA fine tunes the dynamics of a cell state transition involving *Lin28*, *let-7* and *de novo* DNA methylation

Meng Amy Li[1,2]*[†], Paulo P Amaral[3], Priscilla Cheung[2], Jan H Bergmann[4], Masaki Kinoshita[1], Tüzer Kalkan[1], Meryem Ralser[1], Sam Robson[3‡], Ferdinand von Meyenn[5], Maike Paramor[1], Fengtang Yang[6], Caifu Chen[7], Jennifer Nichols[1], David L Spector[4], Tony Kouzarides[3], Lin He[2]*, Austin Smith[1,8]*

[1]Wellcome Trust - Medical Research Council Stem Cell Institute, University of Cambridge, Cambridge, United Kingdom; [2]Division of Cellular and Developmental Biology, Department of Molecular and Cellular Biology, University of California Berkeley, Berkeley, United States; [3]The Gurdon Institute, University of Cambridge, Cambridge, United Kingdom; [4]Cold Spring Harbor Laboratory, Cold Spring Harbor, United States; [5]Babraham Institute, Cambridge, United Kingdom; [6]Wellcome Trust Sanger Institute, Hinxton, United Kingdom; [7]Integrated DNA Technologies, Redwood, United States; [8]Department of Biochemistry, University of Cambridge, Cambridge, United Kingdom

*For correspondence: ml440@
cam.ac.uk (MAL); lhe@berkeley.
edu (LH); austin.smith@cscr.cam.
ac.uk (AS)

Present address: [†]Wellcome
Trust- Medical Research Council
Stem Cell Institute, University of
Cambridge, Cambridge, United
Kingdom; [‡]Bioinformatics Group,
School of Pharmacy &
Biomedical Science, University of
Portsmouth, Portsmouth, United
Kingdom

Competing interests: The
authors declare that no
competing interests exist.

Reviewing editor: Martin Pera,
University of Melbourne,
Australia

**Abstract** Execution of pluripotency requires progression from the naïve status represented by mouse embryonic stem cells (ESCs) to a state capacitated for lineage specification. This transition is coordinated at multiple levels. Non-coding RNAs may contribute to this regulatory orchestra. We identified a rodent-specific long non-coding RNA (lncRNA) *linc1281*, hereafter *Ephemeron* (*Eprn*), that modulates the dynamics of exit from naïve pluripotency. *Eprn* deletion delays the extinction of ESC identity, an effect associated with perduring Nanog expression. In the absence of *Eprn*, *Lin28a* expression is reduced which results in persistence of *let-7 microRNAs, and* the up-regulation of de novo methyltransferases Dnmt3a/b is delayed. *Dnmt3a/b* deletion retards ES cell transition, correlating with delayed *Nanog* promoter methylation and phenocopying loss of *Eprn* or *Lin28a*. The connection from lncRNA to miRNA and DNA methylation facilitates the acute extinction of naïve pluripotency, a pre-requisite for rapid progression from preimplantation epiblast to gastrulation in rodents. *Eprn* illustrates how lncRNAs may introduce species-specific network modulations.

## Introduction

Mouse embryonic stem cells (ESCs), in vitro counterparts of the pre-implantation epiblast, exhibit dual properties of self-renewal and differentiation (*Boroviak et al., 2015*; *Bradley et al., 1984*; *Evans and Kaufman, 1981*; *Martin, 1981*). These properties make them an attractive system for investigating cell fate decision making. In the embryo, spatially and temporally coordinated signals direct the rapid and continuous transition of the epiblast towards lineage specification (*Acampora et al., 2016*; *Smith, 2017*). In contrast, ESCs can be suspended in a ground state of pluripotency, where self-renewal is decoupled from lineage specification, using two inhibitors (2i) of glycogen synthase kinase 3 (GSK3) and mitogen-activated protein kinase kinase (MEK1/2), along with the cytokine leukaemia inhibitory factor (LIF) (*Ying et al., 2008*). Therefore, ESCs provides a unique

experimental system to explore the principles and molecular players underlying the developmental progression of pluripotency (*Kalkan and Smith, 2014*).

While it is increasingly clear that the ESC state is maintained by a core network of transcription factors (*Chen et al., 2008*; *Dunn et al., 2014*; *Ivanova et al., 2006*), less is known about how cells progress from this state to lineage specification (*Buecker et al., 2014*; *Kalkan and Smith, 2014*; *Smith, 2017*). Loss-of-function screens have highlighted a multi-layered machinery that dismantles the naïve state transcription factor network (*Betschinger et al., 2013*; *Leeb et al., 2014*). The latency period for transition depends on the clearance kinetics of network components (*Dunn et al., 2014*). The orchestration of multiple regulators thus ensures rapid and complete dissolution of this core network and consequent timely extinction of ESC identity upon 2i withdrawal (*Kalkan and Smith, 2014*).

In addition to protein coding genes, accumulating evidence suggests that non-coding RNAs can contribute to the regulation of cell fate transitions. Within this class, long non-coding RNAs (lncRNAs) comprise a large fraction of the transcriptome in diverse cell types and exhibit specific spatio-temporal expression (*Carninci et al., 2005*; *Guttman et al., 2009*; *Necsulea et al., 2014*). The genomic distribution of lncRNAs is non-random (*Luo et al., 2016*). A subclass of lncRNAs are divergently transcribed from neighbouring genes and thought to regulate proximal gene expression in cis, either due to the process of transcription (*Ebisuya et al., 2008*; *Engreitz et al., 2016*; *Martens et al., 2004*) or through local lncRNA-protein interactions that recruit regulatory complexes (*Lai et al., 2013*; *Lee, 2012*; *Luo et al., 2016*; *Nagano et al., 2008*). However, the functions and mode of action of the vast majority of lncRNAs remain unknown and require case-by-case experimental determination. In mouse ESCs, knockdowns of a number of lncRNAs have been reported to exert effects on the transcriptome (*Bergmann et al., 2015*; *Dinger et al., 2008*; *Guttman et al., 2011*; *Lin et al., 2014*; *Sheik Mohamed et al., 2010*) and in some cases impair self-renewal (*Lin et al., 2014*; *Luo et al., 2016*; *Savić et al., 2014*).

We investigated the potential involvement of lncRNAs in transition from the naïve ESC state and identified a dynamically regulated lncRNA (linc1281) that we named *Ephemeron* (*Eprn*). We present functional evaluation of *Eprn* and delineation of a downstream genetic interaction network, which is an additional component of the regulatory machinery driving the irreversible and rapid progression from naïve pluripotency in rodent.

## Results

### Identification of lncRNAs associated with transition from naïve pluripotency

Post-implantation epiblast derived stem cells (EpiSCs) represent a primed state of pluripotency developmentally downstream of naïve state ESCs (*Brons et al., 2007*; *Nichols and Smith, 2009*; *Tesar et al., 2007*). To identify lncRNA candidates with a possible role in ESC transition, we analysed in silico the effect of genetic perturbation on expression of ESC and EpiSC states based on published data. We first selected genes that are over ten-fold differentially enriched in ESCs (182 genes) and EpiSCs (131 genes) relative to each other as molecular signatures to represent these two states (*Tesar et al., 2007*). Using published data, we investigated the impact on these two signature sets when individual lncRNAs (147 in total) and known protein coding regulators (40 in total) were knocked down in ESCs grown in LIF/serum (*Guttman et al., 2011*) (*Figure 1A*, *Figure 1—source data 1*). Serum culture supports a heterogeneous mixture of naïve, primed and intermediate cells (*Chambers et al., 2007*; *Kolodziejczyk et al., 2015*; *Marks et al., 2012*). Therefore, analysis in this condition could potentially reveal regulators of the ESC and EpiSC states. The effect of each gene knockdown was plotted based on the percentage of genes significantly altered within ESC and EpiSC signature sets (FDR < 0.05 and fold change >2 or<0.5 over negative control defined by the original study). We validated the approach by analysing the knockdown effects of known ESC self-renewal regulators. As predicted, depletion of factors that maintain the ESC state, such as Stat3, Esrrb, Sox2 and Klf4, led to a decrease in ESC and increase in EpiSC signature (*Figure 1A*), while knockdown of Oct4 gave rise to a decrease in both ESC and EpiSC signatures, consistent with its requirement in both states (*Niwa et al., 2000*; *Osorno et al., 2012*). With this system, we identified

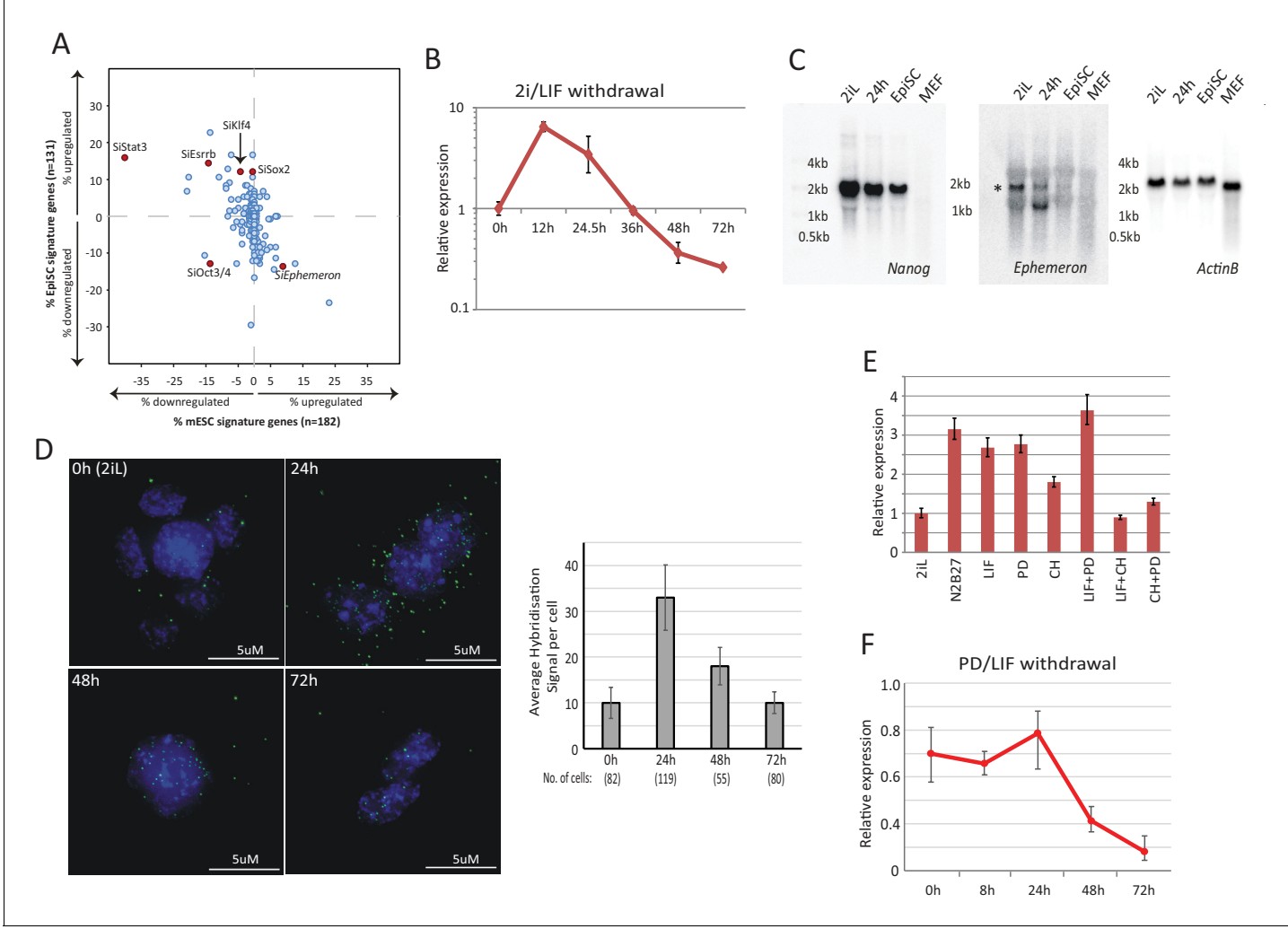

**Figure 1.** Dynamic expression of lncRNA *Ephemeron* during exit from naïve pluripotency. (**A**) Bioinformatic analysis of potential lncRNA candidates in naïve state regulation based on published transcriptome data for lncRNA and pluripotency related gene knockdowns. Each dot represents the effect on ESC (x-axis) and EpiSC (y-axis) gene signatures when a given gene is knocked down. (**B**) RT-qPCR detection of *Eprn* expression relative to *β-actin* upon 2i/LIF withdrawal. Mean ±SD, n = 3. (**C**) Northern blotting of *Eprn*, *Nanog* and *β-actin* in ESCs in 2i/LIF or withdrawn from 2i/LIF for 24 hr, EpiSCs and MEF. * indicates a cross-hybridising RNA species since part of the probe region overlaps with LINE1-L1 and ERVK TEs. (**D**) RNA-FISH for *Eprn* upon 2i/LIF withdrawal with quantification of average hybridisation signals per cell. Mean value of total hybridisation signals for all cells ± SD, n = 2. (**E**) *Eprn* expression relative to *β-actin* upon 2i/LIF component withdrawal quantified by RT-qPCR. Cells cultured in 2i/LIF and were transferred to N2B27 containing indicated single or dual factors for 24 hr. Mean ±SD, n = 3. F, *Eprn* expression relative to *β-actin* upon PD/LIF withdrawal quantified by RT-qPCR. Mean ± SD, n = 3.

The following source data and figure supplements are available for figure 1:

**Source data 1.** Bioinformatics analysis of all lncRNAs and protein coding genes plotted in *Figure 1A*.

**Source data 2.** Expression of potential lncRNA candidates in facilitating naïve state exit.

**Figure supplement 1.** Molecular characterisation of *Ephemeron*.

**Figure supplement 2.** *Eprn* expression and promoter methylation.

lncRNAs that increased ESC and decreased EpiSC signatures when knocked down, suggestive of a possible role in transition from the ESC state (*Figure 1A* bottom right quadrant).

We examined expression profiles of these candidate lncRNAs during exit from self-renewal in defined conditions, exploiting the Rex1::GFP (RGd2) reporter ESC cell line (*Kalkan et al., 2017*; *Wray et al., 2011*) (*Figure 1—source data 2*). Upon 24 hr of 2i withdrawal, Rex1 expression status can discriminate subpopulations of cells with distinct functional properties, with Rex1-GFP high cells corresponding to undifferentiated ESCs and loss of GFP marking extinction of ESC identity (*Kalkan et al., 2017*). Amongst the 16 candidates analysed, *linc1281* (Refseq entry D630045M09Rik) (*Figure 1—figure supplement 1A*) was the third highest expressed lncRNA across all time points. Notably this lncRNA showed a distinctive profile during the first 24 hr, with differential expression observed between Rex1-GFP high and low cells (*Figure 1B*, *Figure 1—figure supplement 1B*). Due to its dynamic and transient expression profile, we designated linc1281 as *Ephemeron* (*Eprn*). Ribosomal profiling analysis indicated that *Eprn* is indeed a non-coding RNA, with the longest predicted open reading frame (80 amino acids) possessing a ribosome release score typical of a non-coding sequence (*Guttman et al., 2013*). *Eprn* is located in a region of high transposable element (TE) content, with its exons comprised of 76.4% annotated TE sequences (including ERV-K, LINE L1, and SINE B2 elements, *Figure 1—figure supplement 1A*). This genomic region exhibits minimal sequence conservation in mammals (*Figure 1—figure supplement 1A*) and we failed to identify any human homologue either within the syntenic region or elsewhere in the human genome. However, a positionally conserved spliced transcript (CA504619) that shares 79% sequence identity to exon 3 of mouse *Eprn* is present within the rat syntenic region (*Figure 1—figure supplement 1C*). Therefore, it is likely that *Eprn* is conserved in rodents over 30 million years since the mouse-rat lineage divergence.

We conducted RT-qPCR, Northern blotting and RNA-FISH to evaluate expression, transcription variants and subcellular localisation of *Eprn* in ESCs. *Eprn* showed strong induction within 12 hr of 2i/LIF withdrawal, but decreased subsequently (*Figure 1B–D*). In EpiSCs or mouse embryonic fibroblasts (MEFs), *Eprn* expression was below the detection limit (*Figure 1C*). Consistent with the UCSC gene annotation, Northern blotting of total ESC RNA confirmed the expression of a single *Eprn* transcript over 1 kb in length (*Figure 1C*, *Figure 1—figure supplement 1D*). Transcription start and end sites of *Eprn* mapped by 5' and 3' RACE were in agreement with the annotation (*Figure 1—figure supplement 1E,F*). After 24 hr of 2i/LIF withdrawal, *Eprn* RNA-FISH hybridisation signals displayed predominantly cytoplasmic localisation, but from 48 hr onwards the remaining signals were mostly in the nucleus (*Figure 1D*).

To explore the regulation of *Eprn*, two inhibitors and LIF were withdrawn singly or dually for 24 hr. In conditions lacking Gsk3 inhibitor CHIRON99021 (CH), *Eprn* was upregulated (*Figure 1E*). When transferred to non-supplemented N2B27 medium from PD/LIF, *Eprn* expression was maintained for 24 hr before declining (*Figure 1F*). The addition of CH to LIF/serum culture reduced *Eprn* expression within 24 hr irrespective of the presence of MEK inhibitor PD0325901 (PD) (*Figure 1—figure supplement 1G,H*). Therefore, *Eprn* is suppressed by CH in self-renewing ESCs.

Through analysis of published data, we found that during early mouse development, *Eprn* expression peaked at E4.5 and was present in both epiblast and primitive endoderm of the mature blastocyst, but absent or low in E5.5 post-implantation epiblast (*Figure 1—figure supplement 2A*) and later stages between E7 and E17 (*Figure 1—figure supplement 2B*). Amongst adult tissues analysed, *Eprn* was only detected in kidney, and at a much lower level than in ESCs. We also observed that *Eprn* expression is restored upon naïve state resetting from EpiSCs (*Guo et al., 2009*; *Yang et al., 2010*) (*Figure 1—figure supplement 2C,D*). We conclude that *Eprn* expression is highly specific to ESCs and the early mouse embryo.

LINE and ERVL-MaLR elements are present within the *Eprn* proximal promoter region (2 kb upstream of TSS) (*Figure 1—figure supplement 1A*). Such repetitive elements gain DNA CpG methylation dramatically during pre- to post-implantation transition (*Smith et al., 2014*). By examining published data from embryos (*Seisenberger et al., 2012*; *Wang et al., 2014*) and ESC progression in vitro (*Kalkan et al., 2017*), we found that CpG methylation gain at the *Eprn* promoter was more extensive in the primed E6.5 epiblast (3% to 80%) than the average changes across all promoters (9% to 35%) or the genome (24% to 70%) (*Figure 1—figure supplement 2E*). In contrast, no major CpG methylation gain at *Eprn* promoter was present 24 hr post 2i withdrawal. These data suggest

that promoter methylation does not initiate *Eprn* repression, but could contribute to maintain silencing in later epiblast.

## Loss of *Ephemeron* delays exit from naïve pluripotency

Initiation of ESC differentiation in defined media upon withdrawal of self-renewal factors recapitulates features of peri-implantation epiblast development (*Kalkan et al., 2017*). The latency of naïve state exit varies, however, according to the starting self-renewal condition (*Dunn et al., 2014*; *Wray et al., 2011*). Higher activity of the core network in PD/LIF compared with 2i results in slower network dissolution, reflected in later onset of RGd2 downregulation (*Dunn et al., 2014*). These two conditions feature different levels of *Eprn* due to CH mediated suppression in 2i (*Figure 1E*). We generated *Eprn* knockout (KO) ESCs via sequential gene targeting (*Figure 2—figure supplement 1*) and examined the phenotype in each condition. In steady state self-renewal, *Eprn* loss did not affect the Rex1-GFP profile in either case (*Figure 2A,B*). Upon transfer to N2B27, *Eprn* KO cells displayed delayed downregulation of GFP compared to parental cells, measured at 24 hr from 2i culture and 40 hr from PD/LIF culture (*Figure 2B*). By 72 hr, however, GFP expression was fully extinguished from either starting condition (*Figure 2—figure supplement 2A*). A transient delay in GFP downregulation in both culture conditions was also evident upon *Eprn* knockdown using siRNAs (*Figure 2—figure supplement 2B*). To assess the effect of *Eprn* depletion functionally, we conducted colony forming assays. Cells maintained in PD/LIF were subjected to 40 hr culture in N2B27 and then replated at clonal density in 2i/LIF to assay the persistence of ES self-renewal potential (*Betschinger et al., 2013*). *Eprn* KO and knockdown cells both gave rise to substantially more undifferentiated colonies than wild type controls (*Figure 2C,D*, *Figure 2—figure supplement 2C*). Considered together, these results indicate that *Eprn* deficiency impairs timely exit from naïve pluripotency.

## Molecular consequences of *Ephemeron* loss

We performed RNA-sequencing and compared the transcriptome of wild type and *Eprn* KO ESCs using three independently targeted KO ESC lines and three subclones of the parental wild type ESCs. Sixteen genes were differentially expressed between wild type and *Eprn* KO cells both in PD/LIF and after 8 hr withdrawal (Benjamini-Hochberg adjusted $p < 0.05$, fold change >1.5 or<0.7) (*Figure 2—figure supplement 2D*) (*Figure 2—figure supplement 2E*). These include *Tcf15*, which has been associated with transition from the naïve state and has an inverse expression pattern compared to naïve pluripotency factors (*Davies et al., 2013*). *Lin28a* was the most differentially expressed gene in the group, with *Eprn* KO cells displaying a twofold reduction in mean expression level (*Figure 2E*). Although Lin28a is commonly considered as a core pluripotency factor, its expression is actually increased when cells transition out of the naïve state in vivo and in vitro (*Boroviak et al., 2015*; *Kalkan et al., 2017*; *Kumar et al., 2014*; *Marks et al., 2012*). Attenuated downregulation of members of the naïve transcription factor network is one explanation for delayed exit from the ESC state (*Kalkan and Smith, 2014*). We hypothesised that Lin28a could be a negative regulator of the network. We examined expression of naïve pluripotency transcription factors in *Eprn* KO cells and found a higher level of *Nanog* mRNA (*Figure 2E*, *Figure 2—figure supplement 2F*). To characterise the profile of naïve pluripotency dissolution further in *Eprn* KO cells, a PD/LIF withdrawal time course was monitored over 24 hr. The two-fold reduction in *Lin28a* mRNA in *Eprn* KO cells was constant throughout this time course (*Figure 2F*). Conversely, *Nanog* transcript and protein levels remained higher at 16 hr and 24 hr respectively (*Figure 2F*, see also 3E,F for protein). Mean *Klf2* transcript levels appeared higher in *Eprn* KO cells, but below statistical significance. Other members of the naïve network showed similar expression profiles in wild type and *Eprn* KO cells (*Figure 2—figure supplement 2F*). Among peri-implantation epiblast markers, upregulation kinetics for *Fgf5* were unchanged in *Eprn* KO cells, but transcripts for *Dnmt3a*, *Dnmt3b* and *Oct6* remained lower from 16 to 24 hr (*Figure 2—figure supplement 2F*). Although not statistically significant, *Otx2* transcripts appeared modestly reduced throughout the time course, which could be related to the elevated expression of Nanog (*Acampora et al., 2016*).

We restored *Eprn* expression in KO cells by inserting the *Eprn* genomic region under control of the human *EF1α* promoter into the deleted locus (*Figure 2—figure supplement 3A,B*). The rescue cells displayed a wild type exit profile as measured by GFP profile 40 hr post PD/LIF withdrawal

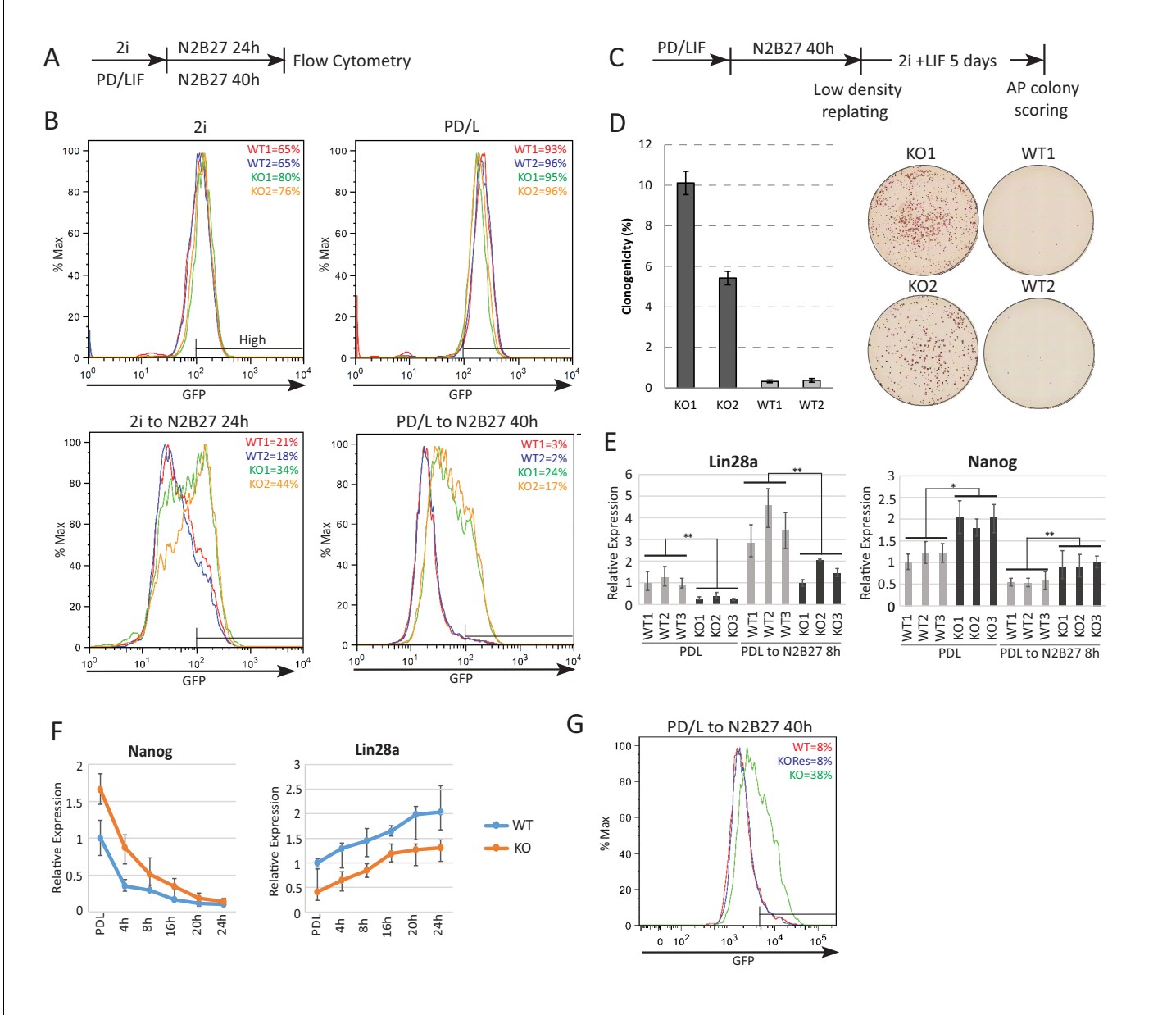

**Figure 2.** Absence of *Ephemeron* delays exit from naïve pluripotency. (**A**) Experimental scheme for analysing naïve state exit using Rex1GFPd2 reporter cells. (**B**) Rex1-GFP flow cytometry profiles of wild type and *Eprn* KO cells in 2i and PD/LIF and during transition from these starting conditions. Two independent clones for wild type and *Eprn* KO cells were analysed. Percentage of GFP high cells were quantified. (**C**) Experimental scheme for colony formation assay. (**D**) Colony formation assay for wild type and *Eprn* KO cells in 2i/LIF 40 hr post PD/LIF withdrawal. Colonies were stained with alkaline phosphatase (AP), with representative images shown. Percentage clonogenicity was calculated by the number of AP positive colonies divided by the total number of cells plated. Mean ± SD, n = 3. (**E**) *Lin28a* and *Nanog* expression relative to *β-actin* in three independent wild type and *Eprn* KO cell lines measured by RT-qPCR. Mean ± SE, n = 3. *p<0.05, **p<0.01, student's *t*-test. (**F**) *Nanog* and *Lin28a* expression kinetics upon PD/LIF withdrawal in wild type and *Eprn* KO cells. Mean ± SD, n = 3. (**G**) Rex1-GFP flow cytometry profile for wild type, *Eprn* KO and *Eprn* rescue cells 40 hr post PD/LIF withdrawal. Percentage of GFP high cells were quantified.

The following figure supplements are available for figure 2:

**Figure supplement 1.** Generation of *Eprn* KO ESCs.

**Figure supplement 2.** Phenotypic and molecular characterisation of *Eprn* KO during naïve state exit.

**Figure supplement 3.** Generation of *Ephemeron* KO rescue ESCs.

*Figure 2 continued on next page*

*Figure 2 continued*

**Figure supplement 4.** Differentiation capacity of *Eprn* KO ESCs.

(*Figure 2G*). *Lin28a* and *Nanog* expression levels were similar to wild type cells (*Figure 2—figure supplement 3C*).

To explore the differentiation capacity of *Eprn* KO cells, we conducted in vitro differentiation assays directing ESCs towards EpiSCs and somatic lineages (*Figure 2—figure supplement 4*). Both wild type and *Eprn* KO ESCs could be differentiated into EpiSCs using N2B27 supplemented with ActivinA/Fgf2/XAV939 (*Sumi et al., 2013*) on fibronectin. Such in vitro differentiated EpiSCs could be stably propagated over multiple passages, and displayed typical morphology and gene expression irrespective of genotype (*Figure 2—figure supplement 4A,B*). We also applied neuronal, mesendoderm and definitive endoderm differentiation protocols to *Eprn* KO ESCs and found that lineage markers were induced, with a slight delay for mesendoderm (*Figure 2—figure supplement 4C–E*). Thus retarded naïve state exit does not notably impair subsequent lineage commitment capacity.

## The *Ephemeron* genetic network includes *Lin28a* and *Nanog*

Based on the preceding data, we hypothesised that Lin28a could be a downstream effector of *Eprn*, acting to reduce expression of Nanog. To characterise further the relationship between *Eprn*, Lin28a and the naïve transcription factor network, we carried out a series of genetic perturbation experiments and measured both Rex1-GFP reporter dynamics and colony formation upon withdrawal from PD/LIF. In *Eprn* KO cells, Nanog knockdown partially restored downregulation of Rex1-GFP 40 hr after PD/LIF withdrawal, and colony formation was reduced to the low level observed in wild type cells subjected to Nanog siRNA (*Figure 3A*). Knockdown of Klf4 had no effect on exit kinetics from PD/LIF in either wild type or *Eprn* KO cells. Knockdowns of other naïve transcription factors, Esrrb, Tfcp2l1 and Klf2, accelerated exit in wild type cells but in contrast to Nanog depletion this phenotype was attenuated in *Eprn* KO cells (*Figure 3—figure supplement 1B*). Resistance of *Eprn* KO cells to accelerated transition upon *Esrrb*, *Tfcp2l1* and *Klf2* knockdown could be attributed to elevated Nanog. We therefore conducted dual knockdown experiments (*Figure 3—figure supplement 1C*). Simultaneous depletion of *Esrrb*, *Tfcp2l1* or *Klf2* together with *Nanog* largely abolished the effect of *Eprn* KO on GFP downregulation (*Figure 3—figure supplement 1C,D*). These data are consistent with *Eprn* acting, at least in part, via modulation of *Nanog* expression.

We investigated whether lowered expression of *Lin28a* contributes to the slower exit from naïve pluripotency and the increased *Nanog* expression. We manipulated *Lin28a* dosage by either overexpression or knockdown in *Eprn* KO cells. In wild type cells, *Lin28a* overexpression had no significant effect. In *Eprn* KO cells, however, it restored normal transition kinetics (*Figure 3B*). Conversely, *Lin28a* knockdown phenocopied *Eprn* loss, delaying exit from naïve pluripotency (*Figure 3C*). Concomitant knockdown of *Nanog* and *Lin28a* abolished this effect (*Figure 3C*). *Lin28a* knockdown cells exhibited marginally elevated *Nanog* mRNA in PD/LIF and persistence at higher levels after 8 hr of PD/LIF withdrawal (*Figure 3D*). At the protein level, *Eprn* null cultures displayed more cells with high Nanog and low Lin28a expression at the 24 hr time point as quantified by co-immunostaining (*Figure 3E,F*, *Figure 3—figure supplement 1E*). Interestingly, Lin28a was detected as concentrated foci in the nucleus and also in the cytoplasm (observed with two independent antibodies), and both nuclear and cytoplasmic expression were increased after PD/LIF withdrawal (*Figure 3—figure supplement 1F*). During early embryo development, expression of *Lin28a* and *Eprn* are positively correlated, while *Lin28a* and *Nanog* are negatively correlated (*Figure 3—figure supplement 1G*) (*Ohnishi et al., 2014*). These data are consistent with the proposition that Lin28a is genetically downstream of *Eprn* and may facilitate exit from naïve pluripotency by accelerating downregulation of Nanog.

To assess whether *Eprn* could regulate *Lin28a* or *Nanog* expression directly, we employed chromatin isolation by RNA purification (ChIRP) (*Chu et al., 2011*). Using this method, we were able to selectively pull down endogenous *Eprn* RNA (*Figure 3—figure supplement 2A*). However, we did

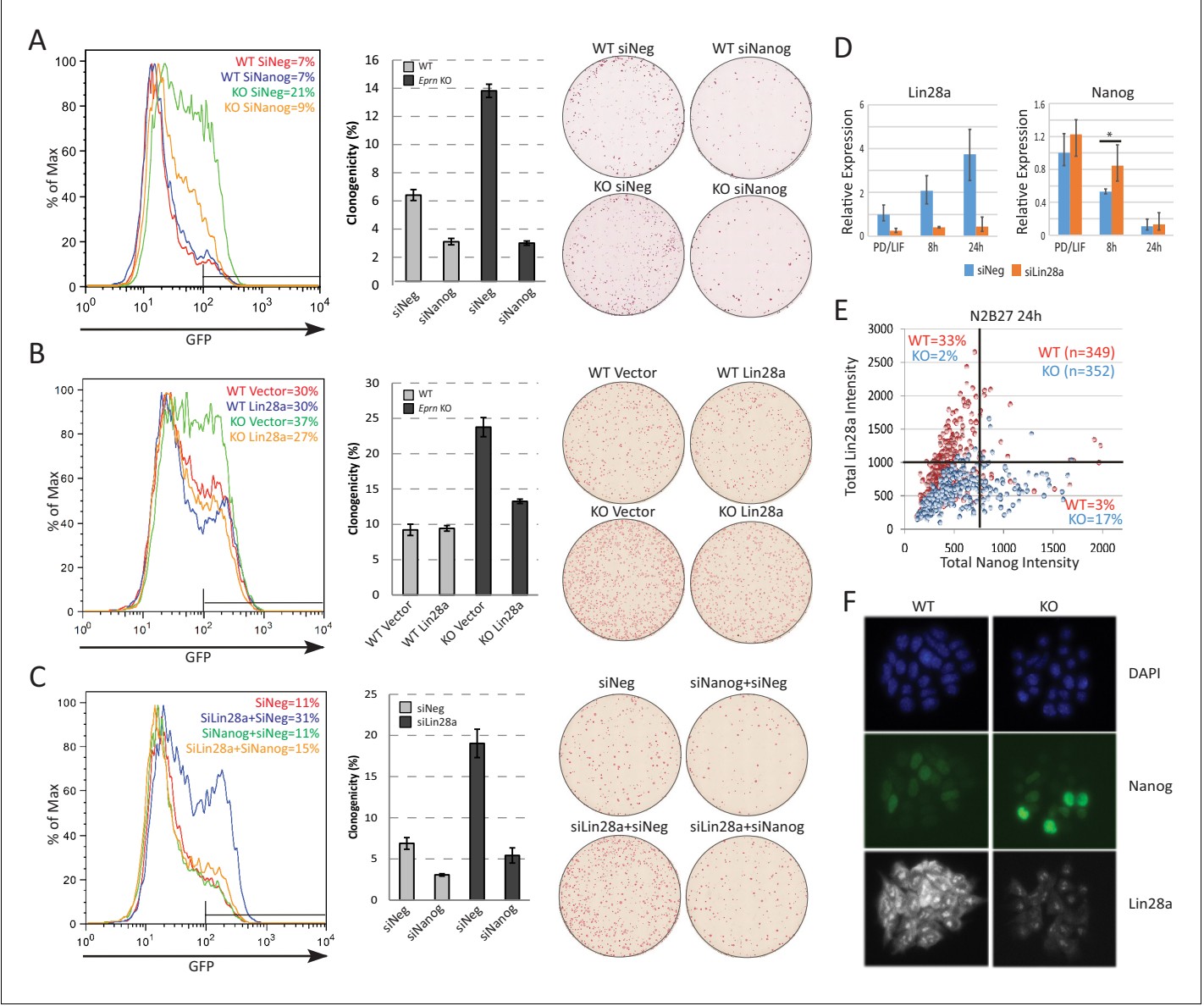

**Figure 3.** Lin28a is downstream of *Ephemeron* and regulates Nanog expression. (**A**) Rex1-GFP flow cytometry profiles (Left) and colony formation capacity (Right) 40 hr post PD/LIF withdrawal for wild type and *Eprn* KO cells transfected with indicated siRNAs. (**B**) Rex1-GFP flow cytometry profiles and colony formation capacity 40 hr post PD/LIF withdrawal for wild type and *Eprn* KO cells transfected with Lin28a expression vector. (**C**) Rex1-GFP flow cytometry profile and colony formation capacity 40 hr post PD/LIF withdrawal with Nanog and Lin28a single or dual knockdowns in wild type cells. Quantification of percentage of GFP high cells were shown in (**A-C**). Percentage clonogenicity in (**A-C**) is measured by the number of AP positive colonies divided by the total number of cells plated, with representative AP staining images shown. Mean ± SD, n = 3. (**D**) *Lin28a* and *Nanog* expression relative to *β-actin* upon PD/LIF withdrawal in Lin28a knockdown and control cells. Mean ± SD, n = 3. *p<0.05, Student's *t*-test. (**E**) Correlation of Nanog and Lin28a protein expression immunostaining in wild type and *Eprn* KO cells 24 hr post PD/LIF withdrawal. (**F**) Representative images of cells co-immunostained with Nanog and Lin28a and quantified in E.

The following figure supplements are available for figure 3:

**Figure supplement 1.** Characterisation of *Eprn*, Nanog and Lin28a genetic interaction.

**Figure supplement 2.** *Eprn* does not act on chromatin.

not detect chromatin enrichment at the *Lin28a* or *Nanog* promoter regions (*Figure 3—figure supplement 2B–D*). Indeed, no significant enrichment genome-wide was observed in wild type compared to *Eprn* KO cells (*Figure 3—figure supplement 2E*). Thus we found no evidence that *Eprn* functions by chromatin association (*Rinn and Guttman, 2014*).

The H3K4me3 modification was reduced at the *Lin28a* promoter in *Eprn* KO cells, in line with reduced transcription (*Figure 3—figure supplement 2F*). One explanation for anti-correlated expression could be direct negative regulation of *Lin28a* by Nanog. We inspected two published Nanog chromatin immunoprecipitation (ChIP) sequencing datasets (*Chen et al., 2008*; *Marson et al., 2008*) but observed no localisation of Nanog at the *Lin28a* locus (*Figure 3—figure supplement 2G*). Furthermore, we did not observe *Lin28a* upregulation in Nanog knockdown cells (*Figure 3—figure supplement 2H*). Therefore, Nanog does not appear to be a direct upstream regulator of Lin28a.

## The function of Lin28a in ESC transition may be mediated by suppression of *let-7*g

Lin28a is an RNA binding protein with a well-established function in suppressing maturation of *let-7* family miRNAs (*Cho et al., 2012*; *Viswanathan et al., 2008*). We investigated whether the role of Lin28a in naïve state exit is *let-7* dependent. We profiled mature miRNA expression of *let-7* family members using RT-qPCR. Expression of *let-7a, let-7d, let-7e, let-7g* and *let-7i* decreased 24 hr after 2i/LIF withdrawal, coincident with the increase in Lin28a expression (*Figure 4A*). Mature miRNA *let-7c* expression was unaffected, suggesting that *let-7c* expression is independent of Lin28a. This observation is in agreement with a recent finding that *let-7c-2*, the major *let-7c* isoform expressed in mouse ESCs, bypasses Lin28a regulation due to lack of a GGAG recognition motif in its loop region (*Triboulet et al., 2015*). The Lin28a regulated *let-7* miRNAs, but not *let-7c*, are expressed at higher levels in ESCs in 2i/LIF than in LIF/serum (*Pandolfini et al., 2016*) (*Figure 4—figure supplement 1A*), consistent with lower Lin28a in 2i/LIF.

To examine the role of Lin28a regulated *let-7* miRNAs in naïve state exit, we first transfected ESCs with mature *let-7g* mimic. We used *let-7g* as a representative member since all apart from *let-7e* share the same seed sequence (*Figure 4—figure supplement 1B*). Forced expression of *let-7g* in RGd2 cells resulted in delayed GFP downregulation upon both 2i and PD/LIF withdrawal (*Figure 4B*). Elevated ESC colony formation capacity post PD/LIF withdrawal was also observed (*Figure 4C*). To identify downstream targets of *let-7g*, we curated genes that are upregulated upon 2i withdrawal in our RNA-sequencing dataset and searched for known or predicted *let-7g* targets using the RNA22 tool (*Miranda et al., 2006*). DNA methyltransferases Dnmt3a and Dnmt3b emerged as prime candidates, as has previously been proposed (*Kumar et al., 2014*). Expression of both increases during ESC transition (*Kalkan et al., 2017*). *Dnmt3a/3b* transcript levels were lower in *Eprn* KO cells than wild type control (*Figure 2—figure supplement 2F*). Multiple *let-7g* target sites were predicted by RNA22 within the *Dnmt3a* 3'UTR and one site in the *Dnmt3b* 3'UTR (*Figure 4D*, *Figure 4—figure supplement 1C*). ESCs were co-transfected with mature *let-7g* mimic and luciferase constructs containing the entire 3'UTRs of *Dnmt3a* and *Dnmt3b*. *let-7g* reduced luciferase expression by more than 60% relative to scrambled control (*Figure 4E*), suggesting that *Dnmt3a/3b* transcripts are indeed *let7-g* targets. To test specificity of this repression, we generated two Dnmt3b 3'UTR luciferase reporter constructs with the *let-7* seed recognition site mutated (*Figure 4—figure supplement 1D*). These mutant reporters escaped repression by the *let-7g* mimic (*Figure 4—figure supplement 1D*).

## Dnmt3a and Dnmt3b methylate the *Nanog* promoter during naïve state exit

Epiblast progression is associated with genome-wide de novo methylation during pre-to post-implantation development (*Auclair et al., 2014*). This phenomenon is recapitulated when naïve ESCs are withdrawn from 2i (*Kalkan et al., 2017*). Previous studies demonstrated hypomethylation of the *Nanog* promoter in mouse ESCs compared to lineage committed cells (*Farthing et al., 2008*; *Yu et al., 2007*). We speculated that impeded de novo DNA methylation could allow perdurance of *Nanog* expression at the onset of naïve state exit. To investigate this hypothesis, we carried out bisulfite sequencing analysis across the *Nanog* proximal promoter region, 1 kb upstream of the TSS,

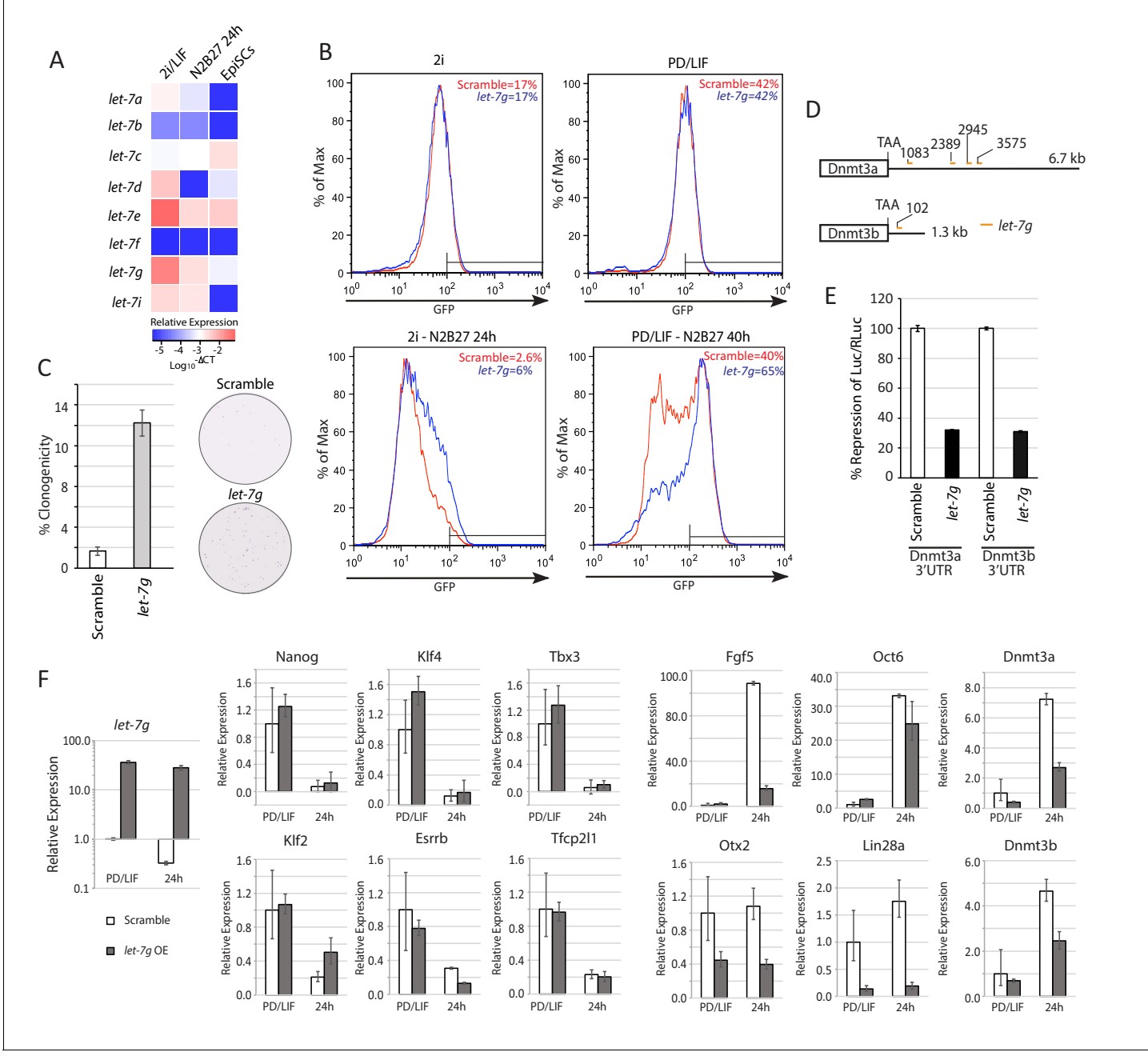

**Figure 4.** Lin28a function is mediated via members of *let-7* miRNAs. (**A**) Mature *let-7* family microRNA expression quantified by RT-qPCR in 2i/LIF, 24 hr post 2i/LIF withdrawal, and EpiSCs. (**B**) Rex1-GFP flow cytometry profile upon forced expression of mature *let-7g* mimic during transition from 2i and PD/LIF. Quantification of percentage of GFP high cells was shown. (**C**) Colony formation assay in 2i/LIF of cells with forced expression of *let-7g* mimic and control 40 hr post PD/LIF withdrawal. Colonies were stained with alkaline phosphatase (AP). Percentage clonogenicity was calculated by the number of AP positive colonies divided by the total number of cells plated. Mean ± SD, n = 3. (**D**) Predicted target sites of *let-7g* in 3′UTRs of *Dnmt3a* and *Dnmt3b* by RNA22. (**E**) Dual luciferase assay measuring repression by *let-7g* mediated through 3′UTRs of *Dnmt3a* and *Dnmt3b*. Fold repression of Luc/Rluc relative to scramble was plotted. Mean ± SD, n = 3. (**F**) Relative expression normalised to *β-actin* of naïve and peri-implantation epiblast associated genes in ESCs with forced expression of *let-7g* mimics. Mean ± SD, n = 3.

The following figure supplement is available for figure 4:

**Figure supplement 1.** *let-7* family mature miRNA expression, sequence and predicted *let-7g* sites in the 3′UTRs of *Dnmt3a* and *Dnmt3b*.

after siRNA knockdown of *Dnmt3a/3b* singly or together (*Figure 5A*). We observed a marked reduction of CpG methylation in the −1 kb to −761 bp region (region 1) 40 hr after PD/LIF withdrawal (*Figure 5A*). Cells transfected with scrambled control siRNA exhibited 40% CpG methylation at scored sites, whereas Dnmt3b depleted cells displayed 18% CpG methylation and Dnmt3a or Dnmt3a/b double knockdown cells showed only around 8%. Effects were less obvious in the −538 bp to +18 bp region (region 2), which was barely methylated at this time point. These data suggest that Dnmt3a and Dnmt3b have overlapping roles in mediating de novo methylation at the *Nanog* proximal promoter. *Eprn* KO cells also exhibited reduced methylation at the Nanog promoter, with region one again showing a more prominent reduction (*Figure 5—figure supplement 1A*).

To explore the role of de novo DNA methylation in ESC transition, we investigated functional consequences of *Dnmt3a* and *Dnmt3b* depletion. We created *Dnmt3a* and *Dnmt3b* single and compound knockouts in RGd2 ESCs using CRISPR/Cas9. Using two guide RNAs (gRNAs), we generated deletions of highly conserved PC and ENV motifs (motifs IV and V) within the catalytic domain for both Dnmt3a and Dnmt3b, recapitulating the previously characterised *Dnmt3b* and *Dnmt3b* mutant gene structures (*Okano et al., 1999*) (*Figure 5—figure supplement 1B*). *Dnmt3a* and *Dnmt3b* single and double KO cells exhibited delayed Rex1-GFP downregulation (*Figure 5B*). Colony formation capacity of the single and double KO cells 40 hr post PD/LIF withdrawal confirmed slower extinction of ESC identity (*Figure 5C*). Interestingly, however, as with *Eprn* KO, the delay in GFP downregulation did not endure (*Figure 5—figure supplement 1C*). Absence of Dnmt3a and Dnmt3b singly or together was associated with transient perdurance of *Nanog* expression (*Figure 5D*). At 8 hr after PD/LIF withdrawal, *Nanog* mRNA in *Dnmt3a* and/or *Dnmt3b* mutants was equivalent to wild type cells in PD/LIF, whereas wild type cells had downregulated *Nanog* expression by 50% (*Figure 5D*). We also observed elevated expression of *Tfcp2l1*, *Klf2*, *Klf4* and *Tbx3* in *Dnmt3a/3b* single or compound KO cells (*Figure 5—figure supplement 1E*). The promoters of these genes are methylation refractory in the 2i withdrawal time course (*Kalkan et al., 2017*). Therefore, the elevated expression should be secondary to some other factor(s) such as increased Nanog. *Dnmt3a/3b* compound KO also resulted in impeded upregulation of peri-implantation markers *Fgf5*, *Oct6* and *Otx2* at 24 hr post PD/LIF withdrawal (*Figure 5—figure supplement 1F*). These data indicate that de novo DNA methylation facilitates timely progression from the ESC state. Importantly, however, methylation by Dnmt3a/3b is not essential for the exit from naïve pluripotency.

## Discussion

Mouse ES cell self-renewal is robust due to recursive wiring of a core transcription factor network (*Dunn et al., 2014*; *Martello and Smith, 2014*; *Young, 2011*). Rapid developmental progression from such a resilient state is achieved through parallel mechanisms. In this study, we find that a lncRNA, *Ephemeron*, participates in the timely dissolution of naïve identity. Genetic interactions link *Eprn* with known players in post-transcriptional and epigenetic regulation (*Figure 5E*). *Eprn* lies upstream of *Lin28a/let-7g* and *Dnmt3a/3b*, and ultimately contributes to timely downregulation of the pivotal naïve transcription factor Nanog. *Eprn* depletion reduces *Lin28a* expression, although the molecular mechanism underlying this effect remains unclear. Lower Lin28a stabilises expression of the *let-7* miRNAs whose targets include de novo DNA methyltransferases Dnmt3a and Dnm3b. Resulting decreased Dnmt3a/3b reduces *Nanog* proximal promoter CpG methylation, correlating with transiently perduring expression. This lncRNA/miRNA/DNA methylation module provides an additional layer in the multi-layered machinery that enforces transition from naïve to formative pluripotency (*Acampora et al., 2016*; *Jang et al., 2017*; *Kalkan et al., 2017*; *Kalkan and Smith, 2014*; *Smith, 2017*).

*Eprn* promotes ESC transition and is suppressed by Gsk3 inhibition during ground state self-renewal in 2i or 2i/LIF. ESCs cultured without Gsk3 inhibition can self-renew in the presence of PD/LIF or LIF/serum. In these conditions they express *Eprn* and higher levels of Lin28a. The lack of overt consequence is presumably due to the dominant self-renewal environment provided by Stat3 activation and MEK inhibition that sustain expression of Nanog and other naïve factors. Nonetheless, loss of *Eprn* in PD/LIF resulted in elevated Nanog and delayed transition kinetics.

We observed a Mendelian ratio of homozygous *Eprn* mutant mice from heterozygous intercrosses (5:18:7, wild type: heterozygous: homozygous offspring). Therefore, in common with *Lin28a* (*Shinoda et al., 2013*), *Eprn* is dispensable for development of laboratory mice. Some protein-

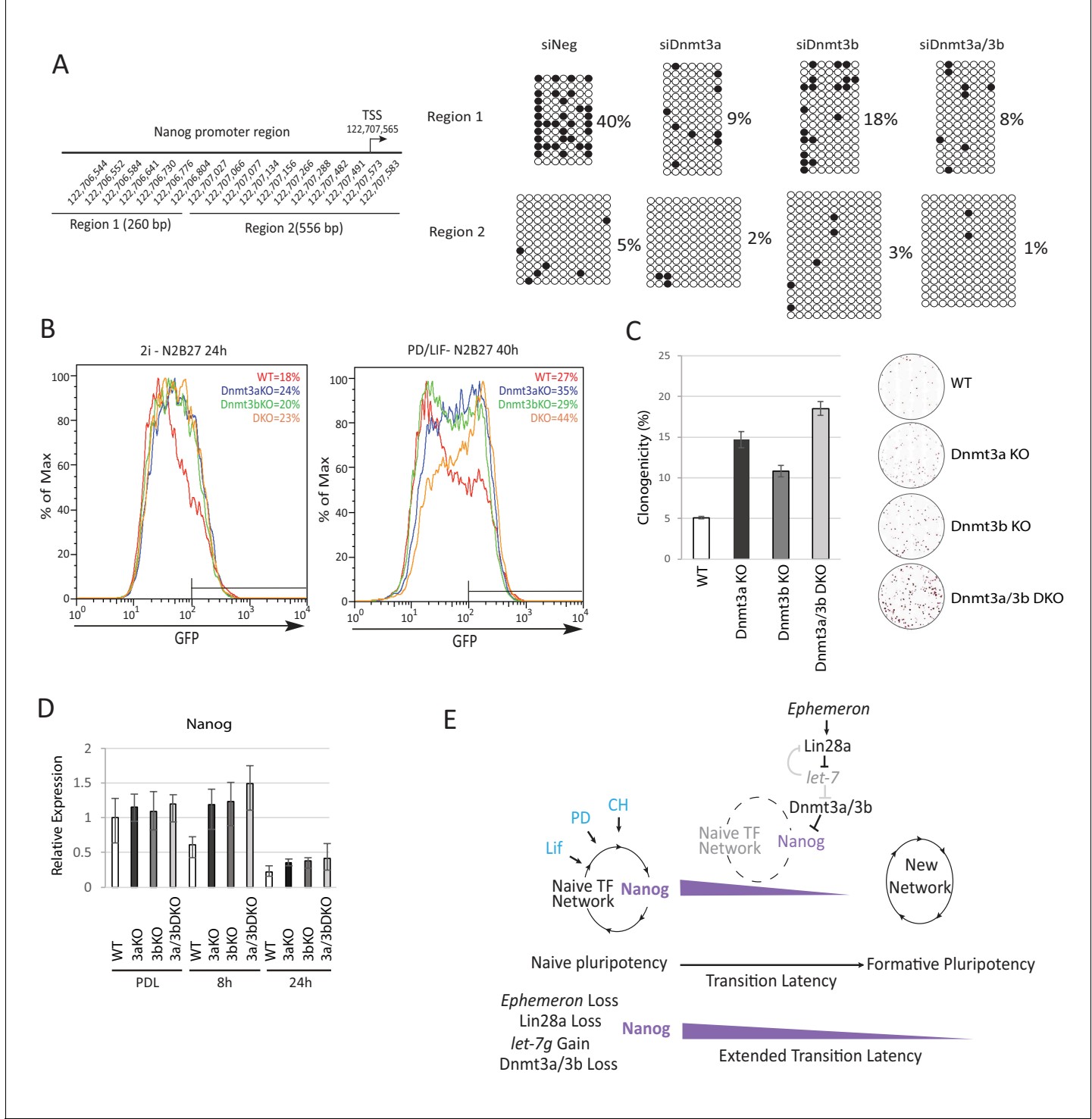

**Figure 5.** Loss of Dnmt3a and Dnmt3b delays naïve state exit associated with transient persistence of Nanog expression. (**A**) Bisulfite sequencing analysis of *Nanog* proximal promoter CpG island DNA methylation in *Dnmt3a* and *Dnm3b* single and dual knockdown cells at 40 hr post PD/LIF withdrawal. The positions of cytosines analysed (mm10) are indicated on the left panel. Black and while circles represent methylated and unmethylated cytosine respectively. (**B**) Rex1-GFP flow cytometry profiles of *Dnmt3a* and *Dnmt3b* single and dual KO cells withdrawn from 2i or PD/LIF for 24 and 40 hr respectively. Percentage of GFP high cells were quantified. (**C**) colony formation capacity 40 hr post PD/LIF withdrawal for *Dnmt3a* and *Dnmt3b* single and compound KO cells. Percentage clonogenicity was measured by the number of AP positive colonies formed divided by the total number of cells plated, with representative AP staining images shown. Mean ± SD, n = 3. (**D**) Expression of *Nanog* relative to *β-actin* in *Dnmt3a* and *Dnmt3b* single

*Figure 5 continued on next page*

*Figure 5 continued*

and compound KO cells quantified by RT-qPCR. Mean ± SD, n = 2. (**E**) Schematic representation of the inferred *Eprn* pathway. Legends for Figures and Source Data.

The following figure supplement is available for figure 5:

**Figure supplement 1.** Phenotypic and molecular characterisation of *Dnmt3a/3b* KO in naïve state exit.

coding genes that exhibit demonstrable loss-of-function phenotypes in ESC self-renewal or transition also show no early embryo phenotype (*Leeb et al., 2014*; *Martello and Smith, 2014*). Our interpretation is that ESCs provide a sensitised platform for identifying components whose functions may be compensated during in vivo development.

The majority of lncRNAs are not phylogenetically conserved (*Necsulea et al., 2014*). Due to their rapidly evolving nature, it is thought that lncRNAs are likely to acquire species or lineage-restricted functions and several examples have recently been described (*Durruthy-Durruthy et al., 2016*; *Paralkar et al., 2014*; *Rani et al., 2016*). The presence of *Eprn* exclusively in rodent may be associated with rapid embryonic progression from pre-implantation epiblast to gastrulation (*Rossant and Tam, 2017*), which necessitates acute extinction of the naïve pluripotency programme (*Smith, 2017*). LncRNAs are more tolerant to TE integration than protein coding genes, which could drive more rapid evolution (*Kelley and Rinn, 2012*; *Necsulea et al., 2014*). Non-coding transcripts harbouring TE sequences are enriched in ESCs and early embryo development in both mouse and human (*Fort et al., 2014*; *Göke et al., 2015*; *Kelley and Rinn, 2012*) and in several instances have been proposed to regulate pluripotency (*Durruthy-Durruthy et al., 2016*; *Fort et al., 2014*; *Lu et al., 2014*). *Eprn* is comprised of 76.4% TEs, compared to the average of 41.4% TE composition in the mouse genome and 33% reported for mouse multi-exon lincRNA sequences (*Kelley and Rinn, 2012*). The aligned sequence between *Eprn* and the rat transcript from the syntenic region includes ERVK LTR and SINE B2 elements. These sequences have been preserved for over 30 million years since the mouse-rat lineage divergence.

Lin28a is known as a human somatic cell reprogramming factor (*Yu et al., 2007*). However, *Lin28a* is expressed at a lower level in ground state mouse ESCs (*Marks et al., 2012*) and pre-implantation epiblast, than in post-implantation epiblast and EpiSCs (*Boroviak et al., 2015*). The expression pattern is consistent with our evidence that up-regulation of *Lin28a* at the onset of mouse ESC differentiation functions to facilitate transition from the naïve state. During human iPSC generation, Lin28a may promote acquisition of primed pluripotency, the endpoint for current human somatic cell reprogramming. Lin28a itself is a target of *let-7* miRNAs (*Kumar et al., 2014*; *Melton et al., 2010*) and the reciprocal negative feedback loops can act as a bimodal switch. Our findings are consistent with the recent report that loss of Lin28a reduced ESC heterogeneity in serum/LIF, favouring a more naïve state (*Kumar et al., 2014*). This effect was attributed to *let-7g*. We note, however, that Lin28a can post-transcriptionally regulate the expression and/or translation of many RNAs independently of *let-7* (*Cho et al., 2012*; *Zhang et al., 2016*) that could also contribute to ESC transition.

De novo methyltransferases Dnmt3a/3b have previously been proposed as targets of *let-7g* (*Kumar et al., 2014*). Our results show that loss of Dnmt3a and Dnmt3b, individually and in combination, delays naïve state exit. Naïve ESCs (*Ficz et al., 2011*; *Habibi et al., 2013*; *Leitch et al., 2013*) and pre-implantation epiblast (*Monk et al., 1987*; *Sanford et al., 1987*) have low expression of Dnmt3a/3b and display global DNA hypomethylation. However, the post-implantation epiblast rapidly acquires global DNA methylation and this process is dependent on Dnmt3a/3b (*Auclair et al., 2014*). A similar acute trend is observed upon naïve ESC withdrawal from 2i (*Kalkan et al., 2017*). Early de novo methylation may have functional consequences for specific naïve pluripotency associated factors, such as Nanog, enduring rapid downregulation. It is noteworthy, however, that the *Dnmt3a/3b* KO phenotype is transient and ESC identity still collapses. Therefore, although de novo DNA methylation facilitates ESC transition it is not mandatory for the exit from naïve pluripotency.

Human naïve pluripotency shares molecular and cellular features with mouse, consistent with a conserved core pluripotency programme in mammals (*Guo et al., 2017*, *2016*; *Smith, 2017*;

*Takashima et al., 2014*; *Theunissen et al., 2014*). However, species-specific differences are evident. Notably, Gsk3 inhibition has less impact on human naïve state maintenance (*Guo et al., 2017*; *Theunissen et al., 2016*). This is partly explained by the lack of *ESRRB* expression in human pluripotent cells (*Blakeley et al., 2015*; *Martello et al., 2012*; *Takashima et al., 2014*; *Theunissen et al., 2014*), but absence of *Eprn* may be an additional factor that reduces requirement for Gsk3 inhibition.

In summary, we have mapped a genetic interaction module consisting of a novel lncRNA, proteins and miRNAs that is integrated into the multi-pronged molecular machinery that propels mouse ESCs towards lineage competence. The fine-tuning effect of *Eprn* could be representative of lncRNA actions in the regulation of molecular networks and illustrative of their potential contribution to species diversification.

## Materials and methods

### Targeting, expression and gRNA vector construction

BAC RP24-353A19 (C57BL/6J background) was obtained from Wellcome Trust Sanger Institute for constructing the *Eprn* knockout targeting vectors by recombineering using bacterial strain EL350 (*Lee et al., 2001*). Floxed drug resistant cassettes containing hygromycin B phosphotransferase gene (Hygro) or Blasticidin S-resistance gene (Bsd) were PCR amplified using chimeric primers miniU and miniD (*Supplementary file 1A*) harbouring 80 bp mini-homologies to the genomic region flanking *Eprn* locus. The purified PCR fragments *loxP*-PGK-Hygro-bghpA-*loxP* and *loxP*-PGK-Bsd-bghpA-loxP were used to replace the entire genomic region of *Eprn* locus with the drug resistant cassettes. The retrieval homology arms were PCR amplified using primers ReUF and ReUR for upstream and ReDF and ReDF for downstream mini-arms (*Supplementary file 1B*). The mini arms were subsequently cloned into pBS-MC1-DTA vector by 3-way ligation using restriction enzymes, SpeI, HindIII and XhoI. The mini-arm containing vector was linearised by HindIII and used to retrieve the targeting vectors from the modified BACs, giving rise to the final targeting vectors, HygroTV and BsdTV.

Lin28a overexpression vector was constructed by PCR cloning mouse *Lin28a* from cDNA using forward primer AATTGTCGACATGGGCTCGGTGTCCAACCAGCAGT and reverse primer AA TTGCGGCCGCTCAATTCTGGGCTTCTGGGAGCAG and cloned into pENTR2B vector. It was subsequently cloned into *PiggyBac*-based expression vector using Gateway LR clonase (Thermo Fisher Scientific, Waltham, MA, USA, 11791020) to generate pCAG-Lin28a-pA:PGK-hygro-pA plasmid.

The gRNA design was conducted using online CRISPR gRNA design tool https://www.atum.bio/eCommerce/cas9/input. The chosen gRNAs were based on minimal off-target scores. Deletions were designed to recapitulate the original Dnmt3a and Dnmt3b KO mutations (*Okano et al., 1999*), excising the highly conserved PC and ENV motifs (motifs IV and V) within the catalytic domain. The gRNAs were generated by annealing the indicated oligos (*Supplementary file 2A*), which were subsequently ligated into pX458 vector (Addgene) digested with BbsI. The constructs were sequence validated before transfection.

### Cell culture

ESCs were cultured on 0.1% gelatin in 2i/LIF medium (homemade N2B27 base medium, supplemented with 1 μM PD0325901, 3 μM CHIR99021, and 20 ng/ml LIF) as described (*Ying et al., 2008*). For gene targeting, ESC were maintained with serum containing medium supplemented with 2i/LIF as above (KO-DMEM high glucose, 15% FCS, 2 mM L-Glutamine, NEAA, 1 mM Sodium Pyruvate (Thermo Fisher Scientific), 100 mM $\beta$-Mercaptoethanol (Sigma Aldrich, St. Louis, MO, USA). Correctly targeted clones were transferred to N2B27 based 2i/LIF medium for expansion and experimentation. The RGd2 reporter wild type subclones and *Eprn* KO ESC clones are of V6.5 origin (RRID:CVCL_C865). An independent wild type RGd2 reporter line is of E14 origin (RRID:CVCL_C320). All cell lines are mycoplasma negative by PCR screening in house.

### Naïve pluripotency exit assays

ESCs were plated at $1 \times 10^4/cm^2$ in 2i without LIF or PD/LIF. The next day, cells were carefully washed with PBS before switching to N2B27 medium. Rex1-GFP profile was analysed at indicated

time points in at least two independent experiments using a Cyan or Fortessa FACs analyser. Live dead discrimination was performed using TO-PRO-3 (Thermo Fisher Scientific, T3605). For clonal assay, post 24 hr or 40 hr 2i or PD/LIF withdrawal respectively, 300–500 cells were plated per well of a 12 well plate coated with Laminin (1:100 dilution, Sigma Aldrich, L2020) and cultured in 2i/LIF for 6 days. Alkaline Phosphatase staining (Sigma Aldrich, 86R-1KT) was conducted to visualise ES colonies. AP-stained plates were imaged using an Olympus IX51, DP72 camera with CellSens software and subsequent colony counting was conducted manually using ImageJ software.

## EpiSC derivation from ESCs and EpiSC resetting

ESCs were plated at $1 \times 10^4$/cm$^2$ in 2i/LIF on a gelatin coated plate. The next day, cells were washed with PBS before medium switch to N2B27 medium supplemented with 20 ng/ml Activin A and 12 ng/ml Fgf2 together with 2 µM XAV939 (Sigma Aldrich, X3004), A/F/X. Cells were then passaged to fibronectin coated plate in A/F/X medium. EpiSCs were passaged for at least seven times before gene expression analysis and resetting. For EpiSC resetting, EpiSCs were stably transfected with GY118F construct by *piggyBac* transposition (*Yang et al., 2010*). $1 \times 10^4$ cells were plated in a one well of a 12 well plate in A/F/X, the next day, 2i plus GCSF was supplied to initiate resetting.

## Differentiation assays

For neuronal differentiation, ESCs were plated at $1 \times 10^4$/cm$^2$ in N2B27 medium on laminin (1:100 in PBS) for up to 4 days for gene expression analysis. For mesendoderm differentiation, cells were plated at $0.6 \times 10^4$/cm$^2$ in N2B27 based medium containing 10 ng/ml ActivinA, 3 µM CHIR99021 on Fibronectin for up to 4 days for gene expression analysis. For definitive endoderm differentiation, cells were plated at $1.5 \times 10^4$/cm$^2$ in N2B27 based medium containing 20 ng/ml ActivinA, 3 µM CHIR99021, 10 ng/ml FGF4, 1 µg/ml Heparin, 100 nM PI103. On day 2, the media was switched to SF5 based medium containing 20 ng/ml ActivinA, 3 µM CHIR99021, 10 ng/ml FGF4, 1 µg/ml Heparin, 100 nM PI103 and 20 ng/ml EGF2. Per 100 ml SP5 basal medium, it contains 500 µl N2, 1 ml B27 without VitaminA supplement, 1% BSA, 1 ml L-glutamine and 100 µl $\beta$-mercaptoethanol. Detailed protocols can be found in Mulas et al (*Mulas et al., 2017*).

## siRNA, miRNA mimics and plasmid transfection

siRNAs and miRNA mimics were obtained from Qiagen and the catalogue numbers are listed in *Supplementary file 3*. Transfection was performed using Dharmafect 1 (Dharmacon, Lafayette, CO, USA, T-2001–01) in a reverse transfection protocol with the final siRNA or miRNA mimics concentration to be 10 nM. Two siRNA combination were used per transfection for each target gene knockdown.

Plasmid transfection was performed using Lipofectamine 2000 (Thermo Fisher Scientific, 11668027) following the manufacturers protocol. For *piggyBac* based stable integration, a *piggyBac* transposon and hyperactive PBase (hyPBase) ratio of 3:1 was used.

## Generation of Dnmt3a and Dnmt3b KO ESCs with CRISPR/Cas9

A pair of gRNA containing plasmids based on px458 backbone (*Ran et al., 2013*) were transfected using Fugene HD (Promega, Madison, WI, USA, E2311). 100 ng of each plasmid were transfected with 0.6 ul Fugene HD (1:3 ratio) to $2 \times 10^5$ ESCs in suspension in 2i/LIF medium overnight. The next day, the media was refreshed and 48 hr post transfection, 1,000 GFP high cells were sorted into a well of a six well plate for colony formation. Individual colonies were picked and genotyping was conducted from extracted genomic DNA by triple primer PCR to identify clones with designed deletion (*Supplementary file 2B*). For Dnmt3a KO, deletion resulted in genotyping PCR product shift from 760 bp representing the wild type allele to 1132 bp. For Dnmt3b KO, deletion resulted in shift from 344 bp representing the wild type allele to 653 bp. Only homozygous mutants were chosen for subsequent experimentation.

## Southern blotting

Genomic DNA of individually picked ESC clones was extracted and digested with *Xmn*I, size-fractionated on a 0.8% agarose gel and transferred to Hybond-XL blotting membrane (GE Healthcare, Chicago, IL, USA, RPN20203) using standard alkaline transfer methods. The 5' and 3' external

probes were generated by PCR with primer sequences shown in *Supplementary file 4A*. Southern blot hybridization was conducted as described previously (*Li et al., 2011*).

## Northern blotting

10 µg of purified RNA was resolved by denaturing formaldehyde agarose gel electrophoresis with MOPS buffer. RNA was transferred to Hybond-XL membrane in 1xSSC buffer overnight by capillary transfer. RNA was UV cross-linked to the membrane and pre-hybridised with Expresshyb (Clone-Tech, Mountainview, CA, USA, 636831) for 2 hr at 65°C. The DNA probe was generated by PCR (primers are shown in *Supplementary file 4B*) and 25 ng of probe DNA was labelled with [$^{32}$P]-dCTP using Radprime DNA labeling system (Thermo Fisher Scientific, 18428–011). The free-nucleotide was removed from labelled probe using G-50 column (GE Healthcare, 27-5330-01), and was heat-denatured followed by snap cooling. The probe was added to the pre-hybridised membrane and incubated overnight at 65°C in a rolling incubator. Membrane was washed with wash buffer containing 0.1 x SSC and 0.1% SDS 3 times at 65°C with 10 min intervals. The membrane was placed in a phosphoimager and exposed for at least overnight at −80°C before scanned using Typhoon 9410 phosphoimager system (GE Healthcare).

## 5' and 3' RACE

5' RACE was conducted using 5'-Full RACE Core Set (Takara, Kusatsu, Japan, #6122) following manufacture's protocol. The sequences for RT-primer and nested PCR primers A1, A2, S1, and S2 are shown in *Supplementary file 5A*. 3' RACE was conducted by using a polyT RT-primer with a unique sequence tag to synthesis cDNA. The 3' end region was PCR amplified using a primer specific to the RT-primer and a gene specific primer. The primers are shown in *Supplementary file 5B*. Both 5' and 3' RACE PCR products were cloned into plasmids using Zero blunt TOPO PCR cloning kit (Thermo Fisher Scientific, 451245) for subsequent sequencing.

## RNA extraction, reverse transcription and Real-time PCR

Total RNA was isolated using Trizol (Thermo Fisher Scientific, 15596026) or RNeasy kit (Qiagen, Hilden, Germany, 74136) and DNase treatment was conducted either after RNA purification or during column purification. cDNA was transcribed from 0.5 ~ 1 ug RNA using SuperScriptIII (Thermo Fisher Scientific, 18080044) and oligo-dT priming. Real-time PCR was performed using StepOnePlus machine (Applied Biosystems) with Fast Sybrgreen master mix (Thermo Fisher Scientific, 4385612). Target gene primer sequences are shown in *Supplementary file 6*. Expression level were normalised to Actinβ. Technical replicates for at least two independent experiments were conducted. The results were shown as mean and standard deviation calculated by StepOnePlus software (Applied Biosystems). The cDNA library for E7-E17 embryos and adult tissues were purchased from Clontech (Mouse Total RNA Master Panel, 636644).

## RNA-FISH

RNA-FISH was conducted using ViewRNA ISH Cell Assay for Fluorescence RNA In Situ Hybridization system (Thermo Fisher Scientific, QVC0001) with modifications and imaged on a DeltaVision Core system (Applied Precision), as described in *Bergmann et al. (2015)*. The probe set used for *Ephemeron* was VX1-99999-01.

## Mature miRNA expression profiling

Total RNA was extracted using Trizol. 1 ug RNA was reverse transcribed using Taqman MicroRNA Reverse Transcription Kit (Thermo Fisher Scientific, 366596). Mature miRNA expression was analysed using Taqman Array Rodent MicroRNA A + B Cared Set V3.0 (Thermo Fisher Scientific, 444909).

## Luciferase assay

The Entire 3'UTR of both *Dnmt3a* and *Dnmt3b* were PCR cloned downstream of the firefly luciferase coding region into pGL3 vector. For *Dnmt3a* 3'UTR, forward primer AATTGGCCGGCCGGGACA TGGGGGCAAACTGAAGTAG and reverse primer AATTGGATCCGGGAAGCCAAAACATAAAGATG TTTATTGAAGCTC were used for PCR cloning. For *Dnmt3b* 3'UTR, forward primer AA TTGGCCGGCCTTCTACCCAGGACTGGGGAGCTCTC and reverse primer AATTGGATCCTTA

TAGAGAAATACAACTTTAATCAACCAGAAAGG were used for PCR cloning. To generate mutant Dnmt3b reporter constructs, *let-7g* binding site was mutated to include BsrGI (mutation V1) and EcoRI (mutation V2) sites by PCR cloning. Each firefly luciferase construct (500 ng) together with Renilla luciferase construct (10 ng) were con-transfected with either *let-7g* mimic or scrambled control (20 nM). The firefly and Renilla luciferase activity was determined by dual luciferase assay (Promega, E1960) 48 hr post-transfection.

## Immunostaining

Cells were fixed in 4% paraformaldehyde for 10 min at room temperature and were blocked with blocking buffer (5% semi-skimmed milk with 0.1% Triton in PBS) for 2 hr at room temperature. Primary antibodies were diluted in blocking buffer and incubated at 4°C overnight. Primary antibody was carefully washed away with 0.1% Triton in PBS three times with 10 min incubation between each wash. Secondary antibody diluted in blocking buffer (1:1000) was incubated at room temperature for 1 hr followed by 3 washes with 0.1% Triton in PBS. Nuclei were counterstained with DAPI. Primary antibodies used were Nanog (eBioscience, 14–5761, RRID:AB_763613, 1:200) and Lin28a (Cell signalling, 3978, RRID:AB_2297060, 1:800; 8706, RRID:AB_10896850, 1:200). Images from random fields were taken with Leica DMI3000 and the images from different fields at each time point were combined and analysed using CellProfiler software (Broad Institute, RRID:SCR_007358) to conduct nuclear and cytoplasmic compartmentalisation and total fluorescent intensity for each sub-cellular compartments as well as the whole cell for each cell was extracted for correlation analysis.

## Chromatin isolation by RNA purification (ChIRP)

The antisense oligo probes were selected with GC content in the range of 40–50% in regions of the *Eprn* exons without repetitive sequences (*Figure 1—figure supplement 1A*). The probes sequences are in shown in *Supplementary file 7*. CHIRP was conducted following published protocol (*Chu et al., 2011*). The data is available at the NCBI Gene Expression Omnibus (accession number: GSE90574). The link to the data is as follows: http://www.ncbi.nlm.nih.gov/geo/query/acc.cgi?acc=GSE90574.

## ChIP

The experimental procedure was conducted as described previously (*Betschinger et al., 2013*). 2 ug of H3K4me3 antibody (Diagenode, Ougrée, Belgium pAb-003–050) and IgG control (Santa Cruz, Dallas, TX, USA, sc-2345) was used for $4 \times 10^6$ cells per ChIP. qPCR was performed with primers shown in *Supplementary file 8*.

## *Nanog* promoter DNA methylation analysis

Genomic DNA was extracted using GenElute Mammalian Genomic DNA miniprep kit (Sigma Aldrich, G1N70-1KT). 500 ng purified genomic DNA was treated with sodium bisulfite to convert all unmethylated cytosine residues into uracil residues using Imprint DNA modification Kit (Sigma Aldrich, MOD50-1KT) according to the manufacturer's protocol. *Nanog* proximal promoter regions (Region 1 and 2 as indicated in *Figure 5a*) were amplified using a nested PCR approach with KAPA HiFi Uracil + Readymix (KapaBiosystems/Roche, Basel, Switzerland, KK2801). The PCR condition for both nested rounds of PCR is as follows: denaturation at 98°C for 5 min followed by 10 cycles of gradient PCR, 98°C for 15 s, 62°C (starting annealing temperature) for 15 s with annealing temperature reduced by 1°C per cycle and 72°C for 1.5 min. Followed by this, a 35 cycles of 98°C for 15 s, 58°C for 15 s and 72°C for 1.5 min were conducted. 2 µl first round PCR product was used as template for the nested PCR. All primer sequences are shown in *Supplementary file 9*. The PCR products were verified and purified by gel electrophoresis and subsequently subcloned by TOPO cloning. Reconstructed plasmids were purified and individual clones were sequenced (Eurofins).

## Transcriptome sequencing and analysis

Total RNA was isolated with RNeasy RNA purification. Ribo-zero rRNA depleted RNA was used to generate sequencing libraries for wild type and Ephemeron knockout cells in PD/LIF and 8 hr withdrawal from PDL from three independent cell lines. Single end sequencing was performed and the reads were mapped using NCBI38/mm10 with Ensembl version 75 annotations. RNA-seq reads were

aligned to the reference genome using tophat2. Only uniquely mapped reads were used for further analysis. Gene counts from SAM files were obtained using htseq-count with mode intersection non-empty, -s reverse. Differential gene expression analysis was conducted using Bioconductor R package DESeq2 version 1.4.5. DESeq2 provides two P-values, a raw P-value and a Benjamini-Hochberg P-value (adjusted p value). An adjusted p-Value threshold of 0.05 was used to determine differential gene expression (95% of the results are not false discoveries, error rate 0.05 = 5%). The data is available at the NCBI Gene Expression Omnibus (accession number: GSE90574, https://www.ncbi.nlm.nih.gov/geo/query/acc.cgi?acc=GSE90574).

### *Eprn* promoter CpG methylation analysis

Using published genome-wide bisulpite sequencing data (*Kalkan et al., 2017*; *Seisenberger et al., 2012*; *Wang et al., 2014*), *Eprn* promoter region was defined as the 2 kb region upstream of the TSS and the percentage of CpG methylation within the region was quantified. For promoter average, percentage of CpG methylation around the 2 kb promoter region of each annotated gene was quantified and averaged for all values. For genome average, percentage of CpG methylation of all 50 kb tiling windows was quantified and averaged all values.

## Acknowledgements

We thank Kosuke Yusa and Graziano Martello for comments on the manuscript. We are grateful to Carla Mulas for assisting the miRNA expression plot, Yiping Zhang for lncRNA candidate prediction analysis and Rosalind Drummond for technical support. We thank Heather Lee for providing Dnmt3a and Dnmt3b siRNAs and Wolf Reik for support. We thank Nicholas Ingolia for useful discussion on lncRNA ribosomal footprinting. We also thank Peter Humphreys and Andy Riddell for technical support for imaging analysis and flow cytometry respectively. AS is supported by Medical Research Council (G1100526/1), Biotechnology and Biological Sciences Research Council (BB/M004023/1), European Commission (HEALTH-F4-2007-200720 EUROSYSTEM), and Wellcome Trust (091484/Z/10/Z). LH is supported by National Cancer Institute (R01 CA139067, 1R21CA175560-01) and California Institute of Regenerative Medicine (RN2-00923-1), American Cancer Society (123339-RSG-12-265-01-RMC), Tobacco-related Disease Research Program (21RT-0133). DLS is supported by NIGMS 42694 and NCI 5PO1CA013106-Project 3. TK is supported by programme grants from Cancer Research UK (C6/A18796) and European Research Council CRIPTON Grant (268569) and core grants from the Wellcome Trust (092096) and Cancer Research UK (C6946/A14492). The Cambridge Stem Cell Institute receives core funding from the Wellcome Trust and the Medical Research Council. MAL was a Siebel postdoctoral fellow at the University of California, Berkeley and a Sir Henry Wellcome postdoctoral fellow (096125/Z/11/Z). AS is a Medical Research Council Professor.

## Additional information

### Funding

| Funder | Grant reference number | Author |
| --- | --- | --- |
| Wellcome | Sir Henry Wellcome Postdoctoral Fellowship | Meng Amy Li |
| California Institute for Regenerative Medicine | Research Grant | Lin He |
| Wellcome | Research Grant | Austin Smith |
| Medical Research Council | Research Grant | Austin Smith |
| National Institute of General Medical Sciences | Research Grant | David L Spector |
| National Cancer Institute | Research Grant | David L Spector Lin He |
| Cancer Research UK | Research Grant | Tony Kouzarides |
| European Research Council | Research Grant | Tony Kouzarides |

| Biotechnology and Biological Sciences Research Council | Research Grant | Austin Smith |
| European Commission | Research Grant | Austin Smith |
| American Cancer Society | Research Scholar Award | Lin He |
| Tobacco-Related Disease Research Program | Research Grant | Lin He |

The funders had no role in study design, data collection and interpretation, or the decision to submit the work for publication.

## Author contributions

MAL, Conceptualization, Data curation, Formal analysis, Funding acquisition, Validation, Investigation, Visualization, Methodology, Writing—original draft, Writing—review and editing; PPA, Formal analysis, Investigation, Methodology, Writing—review and editing; PC, MK, MP, JN, Investigation, Methodology; JHB, FvM, Formal analysis, Investigation, Methodology; TKa, Resources, Data curation; MR, Formal analysis, Visualization; SR, Data curation, Formal analysis, Visualization; FY, Formal analysis, Investigation; CC, Resources, Dr. Chen provided miRNA expression profiling assays; DLS, LH, Supervision, Funding acquisition, Writing—review and editing; TKo, Supervision, Funding acquisition; AS, Supervision, Funding acquisition, Writing—original draft, Writing—review and editing

## Author ORCIDs

Meng Amy Li, http://orcid.org/0000-0002-6619-9919
Austin Smith, http://orcid.org/0000-0002-3029-4682

# Additional files

## Supplementary files

- Supplementary file 1. Primers for generating *Eprn* Targeting vector and genotyping.

- Supplementary file 2. Primers for generating gRNAs vectors and genotyping for *Dnmt3a/3b* knockouts.

- Supplementary file 3. siRNAs and mature miRNA mimics used in this study.

- Supplementary file 4. Primers for generating Southern and Northern blotting probes by PCR.

- Supplementary file 5. Primers for RACE and nested PCR amplification.

- Supplementary file 6. Primers for Real-time quantitative RT-PCR.

- Supplementary file 7. ChIRP probes used in this study.

- Supplementary file 8. Primers used for *Lin28a* promoter ChIP PCR.

- Supplementary file 9. Primers used for *Nanog* promoter DNA methylation analysis.

## Major datasets

The following dataset was generated:

| Author(s) | Year | Dataset title | Dataset URL | Database, license, and accessibility information |
| --- | --- | --- | --- | --- |
| Li MA, Amaral PP, Cheung P, Bergmann JH, Kinoshita M, Kalkan T, Ralser | 2016 | A lncRNA/Lin28/let7 axis coupled to DNA methylation fines tunes the dynamics of a cell state transition | http://www.ncbi.nlm.nih.gov/geo/query/acc.cgi?acc=GSE90574 | Publicly available at the NCBI Gene Expression Omnibus (accession no: |

M, Robson S,
Paramor M, Yang F,
Chen C, Nichols J,
Spector DL, Kou-
zarides T, He L,
Smith A

GSE90574)

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
