## [Decision Letter]

Thank you for submitting your article "A lncRNA/Lin28/let7 axis coupled to DNA methylation fines tunes the dynamics of a cell state transition" for consideration by *eLife*. Your article has been reviewed by three peer reviewers, one of whom is a member of our Board of Reviewing Editors, and the evaluation has been overseen by Fiona Watt as the Senior Editor. The reviewers have opted to remain anonymous.

The reviewers have discussed the reviews with one another and the Reviewing Editor has drafted this decision to help you prepare a revised submission.

Summary:

In this study Li et al. describe the actions of a novel lncRNA in the exit of mouse embryonic stem cells from pluripotency. The authors show that Ephemeron (Eprn), identified through an in silico lncRNA knockdown screen for support of naïve or primed pluripotency, is required for ES cells to progress towards the exit from pluripotency in what the authors refer to as a timely fashion. They present further evidence to show that this effect is mediated through Lin28a via DNMT3a and DNMT3b. The authors further reveal that Eprn is found only in mouse and rat and that its knockout is fully compatible with normal development to term.

The mechanisms that underlie the dismantling of the pluripotent state in preparation for lineage specification are not fully understood. The authors have employed an interesting screening strategy to address this question and have found a lncRNA that certainly seems to play some type of regulatory role in the process.

Essential revisions:

The reviewers have some concerns about the overall significance of the study, and are not convinced regarding the mechanistic studies that link Eprn, Lin28a and downstream targets.

Reviewer 2 notes:

"1) There is really no insight into how the lncRNA functions, 2) The lncRNA is not conserved and thus its relevance appears to be limited to mice, 3) The Eph knockout mouse has no phenotype, 4) along the same lines, the in vitro phenotype is rather subtle, manifesting itself in somewhat contrived experimental conditions."

To clarify the significance of Eprn in the regulation of pluripotency, it would be helpful to provide more information on the knockdown cells.

Reviewer 1:

"Subsection “The function of Lin28a in ESC transition is mediated by suppression of let‐7g” and elsewhere-what exactly do the knockdown cells represent? Are they able to propagate under conditions that support primed pluripotency? This question speaks to how the authors define exit from ESC in these experiments and whether or not this represents the "natural" pathway out of naïve pluripotency."

"The ultimate fate of Eprn KO or knockdown ES cells deprived of self-renewing signals is not clear from the study; although one would assume that they would eventually progress towards germ layer lineage specification, their gene expression patterns do not appear to reflect normal developmental progression of pre-implantation and post-implantation epiblast."

Reviewer 3 notes that the proposed mechanism (Eprn-Lin28a-Nanog) is largely based on correlative evidence. Provide some additional data along the lines suggested by the reviewer concerning Eprn-nanog promoter regulation, Lin28a-nanog promoter interactions and nanog depletion effects on Lin28a.

"Using experiments based on single and combined deficiencies, the authors provde evidence that both Nanog and Lin28a likely act downtream of Eprn, which I find justified. However, I failed to be convinced by their conclusion that Lin28a acts upstream of Nanog in this threesome, and that Eprn directly acts on Lin28a. Accordingly, they could not find enrichment for the Lin28a promoter in Eprn ChIRP experiments. But what about the Nanog promoter in this case? Does Eprn associate with it or not? And could they provide data to document as to whether the promoter of Lin28a is occupied or not by Nanog? Moreover, in all their Nanog depletion experiments, could they report on Lin28a expression?"

[Editors' note: further revisions were requested prior to acceptance, as described below.]

Thank you for resubmitting your work entitled "A lncRNA/Lin28/ Mirlet7 axis coupled to DNA methylation fine tunes the dynamics of a cell state transition" for further consideration at *eLife*. Your revised article has been favorably evaluated by Fiona Watt (Senior editor), a Reviewing editor and two reviewers.

The manuscript has been improved but there are some remaining issues that need to be addressed before acceptance, as outlined below:

You have provided additional information to address many of the concerns raised in the first round of review, as both referees acknowledge, and your reply to the referees was considered and clear. However, both reviewers continue to have concerns about how strongly the evidence supports the molecular regulatory cascade you describe, and about the overall significance of the role of Eprn in control of pluripotency. It would be appropriate to revise the Discussion and the Abstract to acknowledge some of the remaining uncertainties around mechanism and the putative axis of regulation. Concerning the overall significance of the study, your point about species differences in the regulation of transit through pluripotency is important and perhaps should be highlighted a bit more strongly. For example might there be any relationship between the Eprn fine tuning mechanism (and its suppression by GSK3b inhibition) and the ease of capture of naive pluripotency?

Reviewer #2:

This revision adds important additional data and responds well to number of the specific concerns. However, several key concerns remain.

1) Without mechanism, it is unclear whether the Eprn-Lin28 link is direct or in the same pathway. Therefore, describing it as an Eprn-Lin28 axis seems premature.

2) The evidence for *let-7* roles in the phenotype is not convincing. Authors present new analysis of sequencing data showing that while *let-7* is upregulated in the naïve state, it still exists at very low levels (all *let-7* family members combined appear to make up less than 0.3% of all expressed miRNAs). These are low levels especially given that *let-7* function is known to be spread across many targets (1). Also, experiments testing *let-7* function both in the pluripotency phenotype and on suppressing DNMT3a/b are done at non-physiological levels using transfected miRNA mimics. By the authors' measurements, the exogenously introduced *let-7* is introduced at levels at least 30 times higher than the physiological *let-7* levels in the same cells. Therefore, there remains a good chance that the *let-7* results are not physiologically relevant.

3) While I appreciate the importance of the naïve to primed transition as outlined in the response to reviewers, the relatively weak in vitro phenotype (1 day delay) and absence of an in vivo phenotype still raise concern about Eprn's functional importance. Again, I think this issue would not be a major issue if there were more insight into mechanism. That is, either a strong phenotype that provides new major insight into the development of the early mouse embryo or mechanism that provide novel insight to a lncRNA's function would significantly strengthen the paper.

1. A. D. Bosson, J. R. Zamudio, P. A. Sharp. Mol Cell 56, 347-359 (2014).

Reviewer #3:

In this revised version, the authors have provided new data to document the expression patterns of Eprn in vivo and in cellular transitions, the potential for transposable elements and DNA methylation in contributing to Eprn regulation and the epistatic relationship between Eprn, Lin28 and Nanog. They have therefore addressed my request, but we still don' t know how this lnRNA/miRNA/DNA methylation pathway is really organized and acting and it is difficult to estimate whether the Eprn-triggered moleclar cascade has any important role or not.

---

## [Author Response]

*Essential revisions:*

*The reviewers have some concerns about the overall significance of the study, and are not convinced regarding the mechanistic studies that link Eprn, Lin28a and downstream targets.*

We have identified a lncRNA/protein/miRNA pathway which is an integral part of the multi-layered machinery regulating the irreversible exit from the naïve ESC state. Our finding shows that lncRNA *Eprn* has a biological function in fine tuning the dynamics of state transition in pluripotent stem cells. We further delineate a genetic pathway downstream of *Eprn* involving known players in post-transcriptional and epigenetic regulation, specifically Lin28, *let-7* and de novo DNA methyltransferases. This cascade leads to timely silencing of Nanog, a major component of the naïve pluripotency transcription factor network. Our findings have additional value in clarifying the functions of Lin28a/*let-7* and Dnmt3a/3b in embryonic stem cells. Kumar et al., 2014 proposed that Lin28a destabilises the naïve state in a *let-7* dependent fashion and leads to the high heterogeneity of pluripotent stem cells cultured in LIF/serum. Our work unifies these observations with analyses of homogenous stem cell populations in a defined culture system. Specifically, we show that Lin28a acts to promote transition from naïve towards primed pluripotency. This is significant because Lin28 is often presented as a core pluripotency factor. Furthermore, we demonstrated that global de novo methylation, which is rapidly acquired upon naïve state exit in vitro and in vivo (Auclair et al., 2014, Kalkan et al., 2017), facilitates, but is not required for, progression of pluripotency out of the naïve state.

*Reviewer 2 notes:*

1) "There is really no insight into how the lncRNA functions.

Our study aimed to pinpoint a lncRNA that has a measurable biological function and to reveal the downstream genetic interactions. Given the vast number of lncRNAs in the mammalian genome, a key current goals in the field are: (i) to identify biological functions relevant to cellular behaviours, (ii) to elucidate integration with established regulatory frameworks predominantly comprised of protein coding genes. We believe our study is significant in demonstrating substantial progress towards both of these two goals in the context of the previously uncharacterised lncRNA *Eprn*.

Although we do not have a molecular mechanism on how *Eprn* regulate Lin28a expression, we include evidence from the effect of siRNA that *Eprn* RNA is the active entity, and rule out the prevalent mechanism of chromatin association. These results set the stage for investigation of direct molecular mechanism of a lncRNA in a biologically relevant context, but such studies are technologically challenging since there is no consensus on lncRNA mechanism of action and go beyond the scope of our present investigation. We have modified the text to be more explicit about our goals and to qualify our conclusions.

*2) The lncRNA is not conserved and thus its relevance appears to be limited to mice.*

Although *Eprn* is not present in human, it is conserved over 30 million years since mouse-rat divergence. LncRNAs evolve more rapidly than protein coding genes and species–specific lncRNAs represent a large fraction of total lncRNAs (Necsulea et al., 2014). It is thought that lncRNAs are more likely to acquire functions which contribute to species-specific regulation and several examples have recently been discovered (Paralkar et al., 2014, Rani et al., 2016, Durruthy-Durruthy et al., 2016). Understanding how species-specific lncRNAs integrate conserved factors and pathways can offer insight into genome evolution. The rodent specificity of *Eprn* could be directly related to the more rapid progression from the pre-implantation epiblast to gastrulation in rodents than in other mammals, which necessitates acute extinction of the naïve pluripotency program.

3) The Eph knockout mouse has no phenotype.

In vitro culture of ESC provides a sensitised platform for delineating redundant individual components within multi-layered regulatory machineries. We surmise that *Eprn* is dispensable in the laboratory mouse due to the high compensatory capacity of early mouse development. Notably several other well-characterised ES cell regulators, such as Esrrb, Tfcp2l1, LIF, Klf4 and Tfe3, are not essential for the establishment or progression of the epiblast in vivo.

*4) Along the same lines, the* in vitro *phenotype is rather subtle, manifesting itself in somewhat contrived experimental conditions. "*

ESC and naïve pre-implantation epiblast must transit from a naïve state in order to initiate lineage specification. The ESC defined culture system is directly relevant to the trajectory of pre-implantation to early post-implantation epiblast, as documented in recent studies from our group and others (Kalkan et al., 2017; Mulas et al., 2016; Semrau et al. 2016 BioRxiv; Stumpf et al. 2017 BioRxiv; Jang et al., 2017). We exploited this system to demonstrate that from two different starting conditions ESC exhibit a reproducible *Eprn* mutant phenotype. The phenotype is transient and may be considered subtle, but such fine tuning of complex molecular machinery is likely important in evolutionary terms, as indicated by *Eprn* conservation between mice and rats. We have reworded the text to make the description of the culture system more clear and better justified.

*Reviewer 1:*

*"* Subsection “The function of Lin28a in ESC transition is mediated by suppression of let‐7g”*and elsewhere-what exactly do the knockdown cells represent? Are they able to propagate under conditions that support primed pluripotency? This question speaks to how the authors define exit from ESC in these experiments and whether or not this represents the "natural" pathway out of naïve pluripotency."*

Despite a delay in naïve state exit, *Eprn* KO cells downregulate naïve pluripotency associated markers and concomitantly upregulate post-implantation associated markers, as shown in Figure 2—figure supplement 2. This expression profile indicates that after initial delay *Eprn* KO cells do progress towards an early post-implantation epiblast-like state in vitro, as we have recently documented for wild type ESC in these conditions (Kalkan et al. 2017 Development). In response to the reviewer’s question we provide new data showing that *Eprn* KO cells can be converted into primed EpiSC and maintained for at least 10 passages in medium containing Activin/Fgf2/Xav939. They show similar morphology and gene expression to WT ESC derived EpiSCs, as shown in Figure 2—figure supplement 4. Thus the *Eprn* mutant phenotype is transitory leading to the conclusion that the role of Eprn is to expedite normal timely progression of pluripotency but not to influence the trajectory.

*“The ultimate fate of Eprn KO or knockdown ES cells deprived of self-renewing signals is not clear from the study; although one would assume that they would eventually progress towards germ layer lineage specification, their gene expression patterns do not appear to reflect normal developmental progression of pre-implantation and post-implantation epiblast."*

We now demonstrate that *Eprn* KO cells have the capacity to undergo neuronal and mesendodermal lineage specification. These new data are shown in Figure 2—figure supplement 4 and described in the text in subsection “Molecular consequences of *Ephemeron* loss”. In line with the observations above these findings confirm that *Eprn* depletion does not compromise the potency of transitional cells upon naïve state extinction. This is consistent with the development of embryos lacking *Eprn* and is in line with the phenotypes of several other factors that delay exit from the ESC state in vitro without impairing multi-lineage potency (Betschinger et al., 2013; Leeb et al., 2014; Kalkan and Smith 2014).

*Reviewer 3 notes that the proposed mechanism (Eprn-Lin28a-Nanog) is largely based on correlative evidence. Provide some additional data along the lines suggested by the reviewer concerning Eprn-nanog promoter regulation, Lin28a-nanog promoter interactions and nanog depletion effects on Lin28a.*

"Using experiments based on single and combined deficiencies, the authors provde evidence that both Nanog and Lin28a likely act downtream of Eprn, which I find justified. However, I failed to be convinced by their conclusion that Lin28a acts upstream of Nanog in this threesome, and that Eprn directly acts on Lin28a. Accordingly, they could not find enrichment for the Lin28a promoter in Eprn ChIRP experiments. But what about the Nanog promoter in this case? Does Eprn associate with it or not? And could they provide data to document as to whether the promoter of Lin28a is occupied or not by Nanog? Moreover, in all their Nanog depletion experiments, could they report on Lin28a expression?"

As the reviewer points out, our observations are consistent with *Eprn* acting upstream of Lin28a, but the specific mechanism by which *Eprn* regulates Lin28a expression remains unclear. Regarding the possible regulation of Nanog by *Eprn*, we examined Nanog promoter in our ChIRP experiment. There was no *Eprn* peak present at the Nanog locus (Figure 2—figure supplement 2). In fact, from the genome-wide ChIRP analysis, we did not detect any enriched regions in the present of *Eprn*, indicating that *Eprn* does not function by chromatin association.

As for the possibility of Nanog regulation of Lin28a, we examined two published Nanog ChIP-seq datasets (Chen et al., 2008 and Marson et al., 2008) and could not find any Nanog localisation at the *Lin28a* locus, contrasting with prominent peaks at the bona fide target gene *Esrrb* (Figure 2—figure supplement 2). We also examined the expression of Lin28a in *Nanog* knockdown cells and found no reduction in Lin28a expression during the naïve state exit (Figure 2—figure supplement 2). Therefore we conclude that Lin28a is not directly regulated by Nanog.

[Editors' note: further revisions were requested prior to acceptance, as described below.]

*The manuscript has been improved but there are some remaining issues that need to be addressed before acceptance, as outlined below:*

*You have provided additional information to address many of the concerns raised in the first round of review, as both referees acknowledge, and your reply to the referees was considered and clear. However, both reviewers continue to have concerns about how strongly the evidence supports the molecular regulatory cascade you describe, and about the overall significance of the role of Eprn in control of pluripotency. It would be appropriate to revise the Discussion and the Abstract to acknowledge some of the remaining uncertainties around mechanism and the putative axis of regulation. Concerning the overall significance of the study, your point about species differences in the regulation of transit through pluripotency is important and perhaps should be highlighted a bit more strongly. For example might there be any relationship between the Eprn fine tuning mechanism (and its suppression by GSK3b inhibition) and the ease of capture of naive pluripotency?*

We thank the editors and reviewers for their favourable evaluation. We have further revised the Abstract and Discussion and describe the observations as a “connection” and “genetic interaction module” rather than “pathway” to acknowledge the mechanistic uncertainties.

We have incorporated the suggestion to further highlight the role of *Eprn* in species-specific regulation and in this context have discussed the contrasting differences in the requirement of Gsk3 inhibition in maintaining mouse and human naïve pluripotency. While Gsk3 inhibition protects the naïve state in mouse ES cells, it has little impact in human (Theunissen et al., 2016, Guo et al., 2017 in press). This can partly be explained by the lack of ESRRB expression in human pluripotent cells (Blakeley et al., 2015; Martello et al., 2012; Takashima et al., 2014; Theunissen et al., 2014), but our findings suggest that absence of *Eprn* may be an additional factor that reduces requirement for Gsk3 inhibition. *Eprn* action in the modulation of a molecular network exemplifies the potential contribution of lncRNAs to species diversification.

*Reviewer #2:*

*This revision adds important additional data and responds well to number of the specific concerns. However, several key concerns remain.*

*1) Without mechanism, it is unclear whether the Eprn-Lin28 link is direct or in the same pathway. Therefore, describing it as an Eprn-Lin28 axis seems premature.*

We agree with the reviewer that the direct molecular mechanism linking *Eprn* to Lin28a is missing. We now describe this link as a connection between *Eprn* and Lin28a-*mirlet7*-DNA methylation, rather than an axis, and have revised the title, Abstract and Discussion accordingly.

2) The evidence for let-7 roles in the phenotype is not convincing. Authors present new analysis of sequencing data showing that while let-7 is upregulated in the naïve state, it still exists at very low levels (all let-7 family members combined appear to make up less than 0.3% of all expressed miRNAs). These are low levels especially given that let-7 function is known to be spread across many targets (1).

The reviewer rightly points out that all *let-7* family members combined make up only a small fraction of total miRNA expression in ESCs grown in 2i+LIF. However, a small number of highly expressed miRNA clusters dominate the profile (mir-182/96/183, mir-290-295 and mir-30 clusters constitute 20.2%, 18.3% and 6.0% of all expressed miRNAs respectively). The target pool is restricted amongst these clusters. To reflect potential targets of all expressed miRNAs, we summed the expression of miRNAs sharing the same seed sequence as a class (Figure 6). “GAGGUAG”, which is the seed sequence class for all *let-7* miRNAs ranked 39^th^ (out 1558 seed sequence classes in total), i.e. within the top 2.5% of miRNA seed sequences classes in 2i+LIF. This suggests that *let-7* targets are more likely to be regulated than the majority of miRNA targets in ESCs.

Author response image 1.**DOI:**
http://dx.doi.org/10.7554/eLife.23468.028

*Also, experiments testing let-7 function both in the pluripotency phenotype and on suppressing DNMT3a/b are done at non-physiological levels using transfected miRNA mimics. By the authors' measurements, the exogenously introduced let-7 is introduced at levels at least 30 times higher than the physiological let-7 levels in the same cells. Therefore, there remains a good chance that the let-7 results are not physiologically relevant.*

We agree that the expression level of miRNA and the miRNA/target ratio may be functionally significant as described by Bosson et al. (2014). The interaction affinity and the stability of miRNA and target could also affect the outcome. Therefore, the effect of a given miRNA on an individual target has to be investigated experimentally. Currently the field largely relies on overexpression and reporter assays. We have now extended the luciferase reporter analysis (Figure 4) by assaying mutant Dnmt3b 3’UTR reporters. The results presented in Figure 4—figure supplement 1 confirm that repression by let7-g depends on an intact seed sequence.

We attempted to address the reviewer’s concern regarding non-physiological expression levels by using a *let-7*g inhibitor (miRIDIAN miRNA hairpin inhibitor, Dharmacon) to provide orthogonal evidence. The results were inconclusive, however, because we observed an increase in Dnmt3b but a decrease in Dnmt3a expression upon inhibitor treatment. Since the capacity of this let7-g inhibitor to block other let7 family members is unknown, the approach is technically limited.

Functionally, we demonstrated that *Dnmt3a/b* KO cells have a delayed exit phenotype, phenocopying loss of Lin28a and gain of *let-7*g, which is consistent with *Dnmt3a/b* being targets of *let-7*g. These observations are also consistent with published work (Kumar et al., 2014), in which forced expression of *let-7*g and loss of Lin28a sustain the naïve state in ESCs cultured in LIF/serum. Therefore, our work unifies these observations with analyses of homogenous stem cell populations in defined culture systems and further clarifies the role of Lin28a/let7 in pluripotency regulation.

*3) While I appreciate the importance of the naïve to primed transition as outlined in the response to reviewers, the relatively weak* in vitro *phenotype (1 day delay) and absence of an* in vivo *phenotype still raise concern about Eprn's functional importance. Again, I think this issue would not be a major issue if there were more insight into mechanism. That is, either a strong phenotype that provides new major insight into the development of the early mouse embryo or mechanism that provide novel insight to a lncRNA's function would significantly strengthen the paper.*

We agree that the phenotype of *Eprn* is transient and may be considered subtle. We surmise that rodent specific *Eprn*-mediated regulation contributes to rapid extinction of the naïve state, which is more acute in rodents than other mammals. High compensatory capacity in early development may nonetheless render *Eprn* dispensable, at least in the contest of laboratory reared animals.

Given the vast number of lncRNAs with unknown functions, experimental demonstration of the functional relevance of a lncRNA to cellular behaviour and mapping its genetic interactions to known regulatory mechanisms constitute a contribution in the current phase of lncRNA biology when there are limited technologies available to dissect mechanism. Additionally, we have defined functional effects of Lin28a, *let-7* and Dnmt3a/Dnmt3b in pluripotency progression.

1. A. D. Bosson, J. R. Zamudio, P. A. Sharp. Mol Cell 56, 347-359 (2014).

*Reviewer #3:*

*In this revised version, the authors have provided new data to document the expression patterns of Eprn* in vivo *and in cellular transitions, the potential for transposable elements and DNA methylation in contributing to Eprn regulation and the epistatic relationship between Eprn, Lin28 and Nanog. They have therefore addressed my request, but we still don' t know how this lnRNA/miRNA/DNA methylation pathway is really organized and acting and it is difficult to estimate whether the Eprn-triggered moleclar cascade has any important role or not.*

As discussed in our response to reviewer 2 above, although we do not have direct mechanism for *Eprn* regulation of Lin28, we believe that we have provided strong evidence for the genetic interactions between *Eprn, Lin28a/let-7* and *Nanog*. Furthermore, we have provided evidence that Dnmt3a/3b are targets of the *let-7* family and effect *Nanog* promoter methylation during ES cell transition. The *Eprn*/miRNA/Dnmt3a/b interaction module is part of a multi-layered machinery that provides acute extinction of the naïve state in rodents. Eprn exemplifies a role of lcnRNAs in fine-tuning regulatory circuitry in species-specific fashion, which we suggest is an important principle.